# Non-parametric Models for Non-negative Functions

**Ulysse Marteau-Ferey**  **Francis Bach**  **Alessandro Rudi**
INRIA - École Normale Supérieure - PSL Reasearch University
{ulysse.marteau-ferey, francis.bach, alessandro.rudi}@inria.fr

## Abstract

Linear models have shown great effectiveness and flexibility in many fields such as machine learning, signal processing and statistics. They can represent rich spaces of functions while preserving the convexity of the optimization problems where they are used, and are simple to evaluate, differentiate and integrate. However, for modeling non-negative functions, which are crucial for unsupervised learning, density estimation, or non-parametric Bayesian methods, linear models are not applicable directly. Moreover, current state-of-the-art models like generalized linear models either lead to non-convex optimization problems, or cannot be easily integrated. In this paper we provide the first model for non-negative functions which benefits from the same good properties of linear models. In particular, we prove that it admits a representer theorem and provide an efficient dual formulation for convex problems. We study its representation power, showing that the resulting space of functions is strictly richer than that of generalized linear models. Finally we extend the model and the theoretical results to functions with outputs in convex cones. The paper is complemented by an experimental evaluation of the model showing its effectiveness in terms of formulation, algorithmic derivation and practical results on the problems of density estimation, regression with heteroscedastic errors, and multiple quantile regression.

## 1 Introduction

The richness and flexibility of linear models, with the aid of possibly infinite-dimensional feature maps, allowed to achieve great effectiveness from a theoretical, algorithmic, and practical viewpoint in many supervised and unsupervised learning problems, becoming one of the workhorses of statistical machine learning in the past decades [17, 28]. Indeed linear models preserve convexity of the optimization problems where they are used. Moreover they can be evaluated, differentiated and also integrated very easily.

Linear models are adapted to represent functions with unconstrained real-valued or vector-valued outputs. However, in some applications, it is crucial to learn functions with constrained outputs, such as functions which are non-negative or whose outputs are in a convex set, possibly with additional constraints like an integral equal to one, such as in density estimation, regression of multiple quantiles [10], and isotonic regression [5]. Note that the convex pointwise constraints on the outputs of the learned function must hold everywhere and not only on the training points. In this context, other models have been considered, such as generalized linear models [22], at the expense of losing some important properties that hold for linear ones.

In this paper, we make the following contributions:

- We consider a class of models with non-negative outputs, as well as outputs in a chosen convex cone, which exhibit the same key properties of linear models. They can be used within empirical risk minimization with convex risks, preserving convexity. They are defined in terms of an arbitrary feature map and they can be evaluated, differentiated and integrated exactly.

- We derive a representer theorem for our models and provide a convex finite-dimensional dual formulation of the learning problem, depending only on the training examples. Interestingly, in the proposed formulation, the convex pointwise constraints on the outputs of the learned function are naturally converted to convex constraints on the coefficients of the model.

- We prove that the proposed model is a universal approximator and is strictly richer than commonly used generalized linear models. Moreover, we show that its Rademacher complexity is comparable with the one of linear models based on kernels.

- To show the effectiveness of the method in terms of formulation, algorithmic derivation and practical results, we express naturally the problems of density estimation, regression with Gaussian heteroscedastic errors, and multiple quantile regression. We derive the corresponding learning algorithms for convex dual formulation, and compare it with standard techniques used for the specific problems on a few reference simulations.

## 2 Background

In a variety of fields ranging from *supervised learning*, to *Gaussian processes* [36], *inverse problems* [16], *scattered data approximation techniques* [35], and *quadrature methods* to compute multivariate integrals [3], prototypical problems can be cast as

$$f^* \in \arg\min_{f \in \mathcal{F}} \; L(f(x_1), \ldots, f(x_n)) + \Omega(f). \tag{1}$$

Here $L : \mathbb{R}^n \to \mathbb{R}$ is a (often convex) functional, $\mathcal{F}$ a class of real-valued functions, $x_1, \ldots, x_n$ a given set of points in $\mathcal{X}$, and $\Omega$ a suitable regularizer [28].

Linear models for the class of functions $\mathcal{F}$ are particularly suitable to solve such problems. They are classically defined in terms of a feature map $\phi : \mathcal{X} \to \mathcal{H}$ where $\mathcal{X}$ is the input space and $\mathcal{H}$ a separable Hilbert space. Typically, $\mathcal{H} = \mathbb{R}^p$, with $p \in \mathbb{N}$, but $\mathcal{H}$ can also be infinite-dimensional. A linear model is determined by a parameter vector $w \in \mathcal{H}$ as

$$f_w(x) = \phi(x)^\top w, \tag{2}$$

leading to the space $\mathcal{F} = \{f_w \mid w \in \mathcal{H}\}$. These models are particularly effective for problems in the form Eq. (1) because they satisfy the following key properties.

**P1. They preserve convexity of the loss function.** Indeed, given $x_1, \ldots, x_n \in \mathcal{X}$, if $L : \mathbb{R}^n \to \mathbb{R}$ is convex, then $L(f_w(x_1), \ldots, f_w(x_n))$ is convex in $w$.

**P2. They are universal approximators.** Under mild conditions on $\phi$ and $\mathcal{H}$ (universality of the associated kernel function [23]) linear models can approximate any continuous function on $\mathcal{X}$. Moreover they can represent many classes of functions of interest, such as the class of polynomials, analytic functions, smooth functions on subsets of $\mathbb{R}^d$ or on manifolds, or Sobolev spaces [28].

**P3. They admit a finite-dimensional representation.** Indeed, there is a so-called *representer theorem* [15]. Let $L$ be a possibly non-convex functional, $\mathcal{F} = \{f_w \mid w \in \mathcal{H}\}$, and assume $\Omega$ is an increasing function of $w^\top w$ (see [29] for more generality and details). Then, the optimal solution $f^*$ of (1) corresponds to $f^* = f_{w^*}$, with $w^* = \sum_{i=1}^n \alpha_i \phi(x_i)$, and $\alpha_1, \ldots \alpha_n \in \mathbb{R}$. Denoting by $k$ the *kernel function* $k(x, x') := \phi(x)^\top \phi(x')$ for $x, x' \in \mathcal{X}$ (see, e.g., [28]), $f^*$ can be rewritten as

$$f^*(x) = \sum_{i=1}^n \alpha_i k(x, x_i). \tag{3}$$

**P4. They are differentiable/integrable in closed form.** Assume that the kernel $k(x, x')$ is differentiable in the first variable. Then $\nabla_x f_{w^*}(x) = \sum_{i=1}^n \alpha_i \nabla_x k(x, x_i)$. Also the integral of $f_{w^*}$ can be computed in closed form if we know how to integrate $k$. Indeed, for $p : \mathcal{X} \to \mathbb{R}$ integrable, we have $\int f_{w^*}(x) p(x) dx = \sum_{i=1}^n \alpha_i \int k(x, x_i) p(x) dx$.

**Vector-valued models.** By juxtaposing scalar-valued linear models, we obtain a vector valued linear model, i.e. $f_{w_1 \cdots w_p} : \mathcal{X} \to \mathbb{R}^p$ defined as $f_{w_1 \cdots w_p}(x) = (f_{w_1}(x), \ldots, f_{w_p}(x)) \in \mathbb{R}^p$.

### 2.1 Models for non-negative functions or functions with constrained outputs

While linear models provide a powerful formalization for functions from $\mathcal{X}$ to $\mathbb{R}$ or $\mathbb{R}^p$, in some important applications arising in the context of unsupervised learning, non-parametric Bayesian

methods, or graphical models, additional conditions on the model are required. In particular, we will focus on *pointwise output constraints*. That is, given $\mathcal{Y} \subsetneq \mathbb{R}^p$, we want to obtain functions satisfying $f(x) \in \mathcal{Y}$ for all $x \in \mathcal{X}$. A prototypical example is the problem of density estimation.

**Example 1** (density estimation problem)**.** *The goal is to estimate the density of a probability $\rho$ on $\mathcal{X}$, given some i.i.d. samples $x_1, \ldots, x_n$. It can be formalized in terms of Eq. (1) (e.g., through maximum likelihood), with the constraint that $f$ is a density, i.e., $f(x) \geq 0, \;\; \forall x \in \mathcal{X}$, and $\int_{\mathcal{X}} f(x)dx = 1$.*

Despite the similarity with Eq. (1), linear models cannot be applied because of the constraint $f(x) \geq 0$. Existing approaches to deal with the problem above are reported below, but lack some of the crucial properties **P1-4** that make linear models so effective for problems of the form Eq. (1).

**Generalized linear models (GLM).** Given a suitable map $\psi : \mathbb{R}^p \to \mathcal{Y}$, these models are of the form $f(x) = \psi(w^\top \phi(x))$. In the case of non-negative functions, common choices are $\psi(z) = e^z$, leading to the *exponential family*, or the positive part function $\psi(z) = \max(0, z)$. GLM have an expressive power comparable to linear models, being able to represent a wide class of functions, and admit a finite-dimensional representation [14] (they thus satisfy **P2** and **P3**). However, in general they do not preserve convexity of the functionals where they are used (except for specific cases, such as $L = -\sum_{i=1}^n \log z_i$ and $\psi(z) = e^z$ [22]). Moreover they cannot be integrated in closed form, except for specific $\phi$, requiring some Monte Carlo approximations [26] (thus missing **P1** and **P4**). An elegant way to obtain a GLM-like non-negative model is via *non-parametric mirror descent* [37] (see, e.g., their Example 4). A favorable feature of this approach is that the map $\psi$ is built implicitly according to the geometry of $\mathcal{Y}$. However, still the resulting model does not always satisfy **P3**, does not satisfy **P1** and **P4** , and is only efficient in small-dimensional input spaces.

**Non-negative coefficients models (NCM).** Leveraging the finite-dimensional representation of linear models in Eq. (3), the NCM models represent non-negative functions as $f(x) = \sum_{i=1}^n \alpha_i k(x, x_i)$, with $\alpha_1, \ldots \alpha_n \geq 0$, given a kernel $k(x, x') \geq 0$ for any $x, x' \in \mathcal{X}$, such as the Gaussian kernel $e^{-\|x-x'\|^2}$ or the Abel kernel $e^{-\|x-x'\|}$. By construction these models satisfy **P1**, **P3**, **P4**. However, they do not satisfy **P2**. Indeed the fact that $\alpha_1, \ldots, \alpha_n \geq 0$ does not allow cancellation effects and thus strongly constrains the set of functions that can be represented, as illustrated below.

**Example 2.** *The NCM model cannot approximate arbitrarily well a function with a width strictly smaller that the width of the kernel. Take $k(x, x') = e^{-\|x-x'\|^2}$ and try to approximate the function $e^{-\|x\|^2/2}$ on $[-1, 1]$. Independently of the chosen $n$ or the chosen locations of the points $(x_i)_{i=1}^n$, it will not be possible to achieve an error smaller than a fixed constant (Appendix D for a simulation).*

**Partially non-negative linear models (PNM).** A partial solution to have a linear model that is pointwise non-negative is to require non-negativity only on the observed points $(x_i)_{i=1}^n$. That is, the model is of the form $w^\top \phi(x)$, with $w \in \{w \in \mathcal{H} \mid w^\top \phi(x_1) \geq 0, \ldots, w^\top \phi(x_n) \geq 0\}$. While this model is easy to integrate in Eq. (1), this does not guarantee the non-negativity outside of a neighborhood of $(x_i)_{i=1}^n$. It is possible to enrich this construction with a set of points that cover the whole space $\mathcal{X}$ (i.e., a fine grid, if $\mathcal{X} = [-1, 1]^d$), but this usually leads to exponential costs in the dimension of $\mathcal{X}$ and is not feasible when $d \geq 4$.

## 3  Proposed Model for Non-negative Functions

In this section we consider a non-parametric model for non-negative functions and we show that it enjoys the same benefits of linear models. In particular, we prove that it satisfies at the same time all the properties **P1**, ..., **P4**. As linear models, the model we consider has a simple formulation in terms of a feature map $\phi : \mathcal{X} \to \mathcal{H}$. Let $\mathcal{S}(\mathcal{H})$ be the set of bounded Hermitian linear operators from $\mathcal{H}$ to $\mathcal{H}$ (symmetric $p \times p$ matrices if $\mathcal{H} = \mathbb{R}^p$ with $p \in \mathbb{N}$) and denote by $A \succeq 0$ the fact that $A$ is a positive semi-definite operator (a positive semi-definite matrix, when $\mathcal{H}$ is finite-dimensional) [25, 19]. The model is defined for all $x \in \mathcal{X}$ as

$$f_A(x) = \phi(x)^\top A \phi(x), \qquad \text{where} \qquad A \in \mathcal{S}(\mathcal{H}), \;\; A \succeq 0. \tag{4}$$

The proposed model [1] is parametrized in terms of the operator (or matrix when $\mathcal{H}$ is finite dimensional) $A$, like in [9], but with an additional positivity constraint. In particular, this model includes

all the non-negative functions of the form $f_w(x) = (w^\top \phi(x))^2$, $w \in \mathcal{H}$, by taking $A = ww^\top$, i.e., $A \succeq 0$, $\mathrm{rank}(A) \leq 1$. However, while this parameterization $w \mapsto f_w$ is not linear in $w$, the proposed model Eq. (4) is at the same time linear in $A$ (hence leading to convex optimization problems) *and* non-negative for any $x \in \mathcal{X}$, due to the positiveness of the operator $A$, as reported in Proposition 1 below (the complete proof in Appendix B.1).

**Proposition 1** (Pointwise positivity and linearity in the parameters). *Given $A, B \in \mathcal{S}(\mathcal{H})$ and $\alpha, \beta \in \mathbb{R}$, then $f_{\alpha A + \beta B}(x) = \alpha f_A(x) + \beta f_B(x)$. Moreover, $A \succeq 0 \implies f_A(x) \geq 0$, $\forall x \in \mathcal{X}$.*

An important consequence of linearity of $f_A$ in the parameter is that, despite the pointwise non-negativity in $x$, it preserves **P1**, i.e., the convexity of the functional where it is used. First define the set $\mathcal{S}(\mathcal{H})_+$ as $\mathcal{S}(\mathcal{H})_+ = \{A \in \mathcal{S}(\mathcal{H}) \mid A \succeq 0\}$ and note that $\mathcal{S}(\mathcal{H})_+$ is convex [12].

**Proposition 2** (The model satisfies **P1**). *Let $L : \mathbb{R}^n \to \mathbb{R}$ be a jointly convex function and $x_1, \dots, x_n \in \mathcal{X}$. Then the function $A \mapsto L(f_A(x_1), \dots, f_A(x_n))$ is convex on $\mathcal{S}(\mathcal{H})_+$.*

Proposition 2 is proved in Appendix B.2. The property above provides great freedom in choosing the functionals to be optimized with the proposed model. However, when $\mathcal{H}$ has very high dimensionality or it is infinite-dimensional, the resulting optimization problem may be quite expensive. In the next subsection we provide a representer theorem and finite-dimensional representation for our model, that makes the optimization independent from the dimensionality of $\mathcal{H}$. Moreover nevertheless $A$ is a matrix (an operator when $\mathcal{H}$ is infinite dimensional), in Thm. 2 and Remark 1 we will show that the model is not fundamentally over-parametrized compared to a standard linear model.

### 3.1 Finite-dimensional representations, representer theorem, dual formulation

Here we will provide a finite-dimensional representation for the solutions of the following problem,

$$\inf_{A \succeq 0} L(f_A(x_1), \dots, f_A(x_n)) + \Omega(A), \tag{5}$$

given some points $x_1, \dots, x_n \in \mathcal{H}$. However, the existence and uniqueness of solutions for the problem above depend crucially on the choice of the regularizer $\Omega$ as it happens for linear models when $\mathcal{H}$ is finite-dimensional [16]. To derive a representer theorem for our model, we need to specify the class of regularizers we are considering. In the context of linear models a typical regularizer is Tikhonov regularization, i.e., $\Omega(w) = \lambda w^\top w$, for $w \in \mathcal{H}$. Since the proposed model is expressed in terms of a symmetric operator (matrix, if $\mathcal{H}$ is finite-dimensional), the equivalent of the Tikhonov regularizer is a functional that penalizes the squared Frobenius norm of $A$, i.e., $\Omega(A) = \lambda \mathrm{Tr}(A^\top A)$, for $A \in \mathcal{S}(\mathcal{H})$ also written as $\Omega(A) = \lambda \|A\|_F^2$ [16]. However, since $A$ is an operator, we can also consider different norms on its spectrum. From this viewpoint, an interesting regularizer corresponds to the *nuclear norm* $\|A\|_\star$, which induces sparsity on the spectrum of $A$, leading to low-rank solutions [24, 9]. In this paper, for the sake of simplicity we will present the results for the following regularizer, which is the matrix/operator equivalent of the *elastic-net* regularizer [39]:

$$\Omega(A) = \lambda_1 \|A\|_\star + \lambda_2 \|A\|_F^2, \quad \forall A \in \mathcal{S}(\mathcal{H}), \tag{6}$$

with $\lambda_1, \lambda_2 \geq 0$ and $\lambda_1 + \lambda_2 > 0$. Since elastic-net penalization induces low-rank solutions [39] when $\lambda_1 > 0$, this choice allows to obtain models similar to $f_w(x) = (w^\top \phi(x))^2$ mentioned above, but preserving the convexity of the problem. Note that $\Omega$ is strongly convex as soon as $\lambda_2 > 0$; we will therefore take $\lambda_2 > 0$ in practice in order to have easier optimization. Recall the definition of the kernel $k(x, x') := \phi(x)^\top \phi(x')$, $x, x' \in \mathcal{X}$ [28]. We have the following theorem.

**Theorem 1** (Representer theorem, **P3**). *Let $L : \mathbb{R}^n \to \mathbb{R} \cup \{+\infty\}$ be lower semi-continuous and bounded below, and $\Omega$ as in Eq. (6). Then Eq. (5) has a solution $A_*$ which can be written as*

$$A_* = \sum_{i,j=1}^n \mathbf{B}_{ij} \phi(x_i) \phi(x_j)^\top, \quad \text{for some matrix } \mathbf{B} \in \mathbb{R}^{n \times n}, \ \mathbf{B} \succeq 0. \tag{7}$$

*$A_*$ is unique if $L$ is convex and $\lambda_2 > 0$. By Eq. (4), $A_*$ corresponds to a function of the form*

$$f_*(x) := f_{A_*}(x) = \sum_{i,j=1}^n \mathbf{B}_{ij} k(x, x_i) k(x, x_j), \quad \text{for some matrix } \mathbf{B} \in \mathbb{R}^{n \times n}, \ \mathbf{B} \succeq 0.$$

the optimization algorithm is based on an incorrect representation and inefficient at best. See Appendix F for details. This model has also been considered in the literature on sum of squares optimization (see chapter 3 of [8]), which relies on algebraic considerations in the case where $\phi$ is polynomial and $L$ is a linear function.

The proof of the theorem above is in Appendix B.3, where it is derived for the more general class of spectral regularizers (this thus extends a result from [1], from linear operators between potentially different spaces to positive self-adjoint operators).

**Equivalent finite-dimensional formulation in the primal.** A direct consequence of Thm. 1 is the following finite-dimensional representation of the optimization problem in Eq. (5). Denote by $\mathbf{K} \in \mathbb{R}^{n \times n}$ the matrix $\mathbf{K}_{i,j} = k(x_i, x_j)$ and assume w.l.o.g. that it is full rank (always true when the $n$ observations are distinct and $k$ is a *universal kernel* such as the Gaussian kernel [23]). Let $\mathbf{V}$ be the Cholesky decomposition of $\mathbf{K}$, i.e., $\mathbf{K} = \mathbf{V}^\top \mathbf{V}$. Define the finite dimensional model

$$\tilde{f}_{\mathbf{A}}(x) = \Phi(x)^\top \mathbf{A} \Phi(x), \qquad \mathbf{A} \in \mathbb{R}^{n \times n}, \ \mathbf{A} \succeq 0, \tag{8}$$

where $\Phi : \mathcal{X} \to \mathbb{R}^n$, defined as $\Phi(x) = \mathbf{V}^{-\top} v(x)$, with $v(x) = (k(x, x_i))_{i=1}^n \in \mathbb{R}^n$, is the classical *empirical feature map*. In particular, $\tilde{f}_{\mathbf{A}} = f_A$ where $A$ is of the form Eq. (7) with $\mathbf{B} = \mathbf{V}^{-1} \mathbf{A} \mathbf{V}^{-\top}$. We will say that $\tilde{f}_{\mathbf{A}}$ is a solution of Eq. (5) if the corresponding $A$ is a solution of Eq. (5).

**Proposition 3** (Equivalent finite-dimensional formulation in the primal). *Under the assumptions of Thm. 1, the following problem has at least one solution, which is unique if $\lambda_2 > 0$ and $L$ is convex :*

$$\min_{\mathbf{A} \in \mathcal{S}(\mathbb{R}^n), \mathbf{A} \succeq 0} L(\tilde{f}_{\mathbf{A}}(x_1), \dots, \tilde{f}_{\mathbf{A}}(x_n)) + \Omega(\mathbf{A}). \tag{9}$$

*Moreover, for any given solution $\mathbf{A}^* \in \mathbb{R}^{n \times n}$ of Eq. (9), the function $\tilde{f}_{\mathbf{A}^*}$ is a minimizer of Eq. (5). Finally, note that problems Eq. (5) and Eq. (9) have the same condition number if it exists.*

The proposition above (proof in Appendix B.4) characterizes the possibly infinite-dimensional optimization problem of Eq. (5) in terms of an optimization on $n \times n$ matrices. A crucial property is that the formulation in Eq. (9) *preserves convexity*, i.e., it is convex as soon as $L$ is convex. To conclude, Appendix B.4 provides a construction for $\mathbf{V}$ valid for possibly rank-deficient $\mathbf{K}$. We now provide a finer characterization in terms of a dual formulation on only $n$ variables.

**Convex dual formulation.** We have seen above that the problem in Eq. (5) admits a finite-dimensional representation and can be cast in terms of an equivalent problem on $n \times n$ matrices. Here, when $L$ is convex, we refine the analysis and provide a dual optimization problem on only $n$ variables. The dual formulation is particularly suitable when $L$ is a sum of functions as we will see later. In the following theorem $[\mathbf{A}]_-$ corresponds to the negative part[2] of $\mathbf{A} \in \mathcal{S}(\mathbb{R}^n)$.

**Theorem 2** (Convex dual problem). *Assume $L$ is convex, lower semi-continuous and bounded below. Assume $\Omega$ is of the form Eq. (6) with $\lambda_2 > 0$. Assume that the problem has at least a strictly feasible point, i.e., there exists $A_0 \succeq 0$ such that $L$ is continuous in $(f_{A_0}(x_1), ..., f_{A_0}(x_n)) \in \mathbb{R}^n$ (this condition is satisfied in simple cases; see examples in Appendix B.5). Denoting with $L^*$ the Fenchel conjugate of $L$ (see [12]), problem Eq. (9) has the following dual formulation:*

$$\sup_{\alpha \in \mathbb{R}^n} -L^*(\alpha) - \frac{1}{2\lambda_2} \|[\mathbf{V} \operatorname{Diag}(\alpha) \mathbf{V}^\top + \lambda_1 \mathbf{I}]_-\|_F^2, \tag{10}$$

*and this supremum is attained. Moreover, if $\alpha^* \in \mathbb{R}^n$ is a solution of (10), a solution of (5) is obtained via (7), with $\mathbf{B} \in \mathbb{R}^{n \times n}$ defined as*

$$\mathbf{B} = \lambda_2^{-1} \mathbf{V}^{-1} [\mathbf{V} \operatorname{Diag}(\alpha^*) \mathbf{V}^\top + \lambda_1 \mathbf{I}]_- \mathbf{V}^{-\top}. \tag{11}$$

The previous theorem (proof in Appendix B.5) has two main consequences explored below.

**Remark 1** (Same degrees of freedom as linear models). *Thm. 2 shows that Eq. (5) can be cast as a problem with $n$ degrees of freedom: the same number of a standard non-parametric linear models of the form $w^\top \phi(x)$ when a representer theorem is applicable. This is more clear from Eq. (11) where, nevertheless the matrix $\mathbf{B}$ is of dimension $n \times n$, it is completely parametrized by only $n$ variables, showing that the proposed model is not overparametrized with respect to a linear model.*

Moreover, in terms of computations and complexity, the result above is particularly interesting when $L$ can be written in terms of a sum of functions, i.e., $L(z_1, \dots, z_n) = \sum_{i=1}^n \ell_i(z_i)$ for some functions

$\ell_i : \mathbb{R} \to \mathbb{R}$. Then the Fenchel dual is $L^*(\alpha) = \sum_{i=1}^n \ell_i^*(\alpha_i)$, where $\ell_i^*$ is the Fenchel dual of $\ell_i$, and the optimization can be carried by using accelerated proximal splitting methods as FISTA [6], since $\|[\mathbf{V} \operatorname{Diag}(\alpha)\mathbf{V}^\top + \lambda_1 \mathbf{I}]_-\|_F^2$ is differentiable in $\alpha$. As discussed in the appendix this would amount to $O(n^3)$ per iteration for FISTA, due to the computation of Eq. (11), for a total cost of $O(n^3 \epsilon^{-1/2})$ to achieve a solution of error $\epsilon$. Note that $O(n^3)$ can be further reduced and made comparable with fast algorithms for linear models based on kernels [27] (which require $n^2$ or $n\sqrt{n}$), by using techniques from randomized linear algebra and Nyström approximation [18] (see more details in Appendix B.5).

## 4 Approximation Properties of the Model

The goal of this section is to study the approximation properties of our model and to understand its "richness", i.e., which functions it can represent. In particular, we will prove that, under mild assumptions on $\phi$, (a) the proposed model satisfies the property **P2**, i.e., it is a *universal approximator* for non-negative functions, and (b) that it is strictly richer than the family of exponential models with the same $\phi$. First, define the set of functions belonging to our model

$$\mathcal{F}_\phi^\circ = \{ f_A \mid A \in \mathcal{S}(\mathcal{H}), \ A \succeq 0, \|A\|_\circ < \infty \},$$

where $\|\cdot\|_\circ$ is a suitable norm for $\mathcal{S}(\mathcal{H})$. In particular, norms that we have seen to be relevant in the context of optimization are the nuclear norm $\|\cdot\|_\star$ and the Hilbert-Schmidt (Frobenius) norm $\|\cdot\|_F$. Given norms $\|\cdot\|_a, \|\cdot\|_b$, we denote the fact that $\|\cdot\|_a$ is stronger (or equivalent) than $\|\cdot\|_b$ with $\|\cdot\|_a \trianglerighteq \|\cdot\|_b$ (for example, $\|\cdot\|_\star \trianglerighteq \|\cdot\|_F$). In the next theorem we prove that when the feature map is universal [23], such as the one associated to the Gaussian kernel $k(x, x') = \exp(-\|x - x'\|^2)$ or the Abel kernel $k(x, x') = \exp(-\|x - x'\|)$, then the proposed model is a universal approximator for non-negative functions over $\mathcal{X}$ (in particular, in the sense of *cc-universality* [23, 31], see Appendix B.6 for more details and the proof).

**Theorem 3** (Universality, **P2**). *Let $\mathcal{H}$ be a separable Hilbert space, $\phi : \mathcal{X} \to \mathcal{H}$ a universal map [23], and $\|\cdot\|_\star \trianglerighteq \|\cdot\|_\circ$. Then $\mathcal{F}_\phi^\circ$ is a universal approximator of non-negative functions over $\mathcal{X}$.*

The fact that the proposed model can approximate arbitrarily well any non-negative function on $\mathcal{X}$, when $\phi$ is universal, makes it a suitable candidate in the context of nonparametric approximation/interpolation or learning [35, 34] of non-negative functions. In the following theorem, we give a more precise characterization of the functions contained in $\mathcal{F}_\phi^\circ$. Denote by $\mathcal{G}_\phi$ the set of linear models induced by $\phi$, i.e., $\mathcal{G}_\phi = \{ w^\top \phi(\cdot) \mid w \in \mathcal{H} \}$ and by $\mathcal{E}_\phi$ the set of *exponential models* induced by $\phi$,

$$\mathcal{E}_\phi = \{ e^f \mid f(\cdot) = w^\top \phi(\cdot), \ w \in \mathcal{H} \}.$$

**Theorem 4** ($\mathcal{F}_\phi^\circ$ strictly richer than the exponential model). *Let $\|\cdot\|_\star \trianglerighteq \|\cdot\|_\circ$. Let $\mathcal{X} = [-R, R]^d$, with $R > 0$. Let $\phi$ be such that $W_2^m(\mathcal{X}) = \mathcal{G}_\phi$, for some $m > 0$, where $W_2^m(\mathcal{X})$ is the* Sobolev space *of smoothness $m$ [2]. Let $x_0 \in \mathcal{X}$. The following hold:*

*(a) $\mathcal{E}_\phi \subsetneq \mathcal{F}_\phi^\circ$;  (b) the function $f_{x_0}(x) = e^{-\|x - x_0\|^{-2}} \in C^\infty(\mathcal{X})$ satisfies $f_{x_0} \in \mathcal{F}_\phi^\circ$ and $f_{x_0} \notin \mathcal{E}_\phi$.*

Thm. 4 shows that if $\phi$ is rich enough, then the space of exponential models is strictly contained in the space of functions associated to the proposed model. In particular, the proposed model can represent functions that are exactly zero on some subset of $\mathcal{X}$ as showed by the example $f_{x_0}$ in Thm. 4, while the exponential model can represent only strictly positive functions, by construction. Discussion on the condition $W_2^m(\mathcal{X}) = \mathcal{G}_\phi$, proof of Thm. 4 and its generalization to $\mathcal{X} \subseteq \mathbb{R}^d$ are in App. B.7. Here we note only that the condition $W_2^m(\mathcal{X}) = \mathcal{G}_\phi$ is quite mild and satisfied by many kernels such as the Abel kernel $k(x, x') = \exp(-\|x - x'\|)$ [35, 7]. We conclude with a bound on the *Rademacher complexity* [11] of $\mathcal{F}_\phi^\circ$, which is a classical component for deriving generalization bounds [30]. Define $\mathcal{F}_{\phi,L}^\circ = \{ f_A \mid A \succeq 0, \|A\|_\circ \leq L \}$, for $L > 0$. Thm. 5 shows that the Rademacher complexity of $\mathcal{F}_{\phi,L}^\circ$ depends on $L$ and not on the dimensionality of $\mathcal{X}$, as for regular kernel methods [11].

**Theorem 5** (Rademacher compl. of $\mathcal{F}_\phi^\circ$). *Let $\|\cdot\|_\circ \trianglerighteq \|\cdot\|_F$ and $\sup_{x \in \mathcal{X}} \|\phi(x)\| \leq c < \infty$. Let $(x_i)_{i=1}^n$ be i.i.d. samples, $L \geq 0$. The Rademacher complexity of $\mathcal{F}_{\phi,L}^\circ$ is upper bounded by $\frac{2Lc^2}{\sqrt{n}}$.*

The result above (more details and proof in Appendix B.8) shows that also from a statistical viewpoint the proposed model has the same degrees of freedom of a linear model with the same feature map [30].

In particular, by using the result above we can directly derive explicit learning rates, that are in the same order of the one for linear models, as follows. Let $R(f) = \mathbb{E}_{x,y}\ell(y, f(x))$ be the population risk for some $G$-Lipschitz loss function $\ell$ and $\widehat{R}_D$ be the empirical version $\widehat{R}_D(f) = \frac{1}{n}\sum_{i=1}^n \ell(y_i, f(x_i))$ for a given dataset $D$ of $n$ examples. Keeping the same notations as in the theorem, denote by $\widehat{f}_{D,L} = \arg\min_{f \in \mathcal{F}_{\phi,L}^\circ} \widehat{R}_D(f)$ the empirical risk minimization solution over the set $\mathcal{F}_{\phi,L}^\circ$. The *excess risk* over the set $\mathcal{F}_{\phi,L}^\circ$ is bounded as (see Appendix B.9 for more details)

$$\mathbb{E}_D R(\widehat{f}_{D,L}) - \inf_{f \in \mathcal{F}_{\phi,L}^\circ} R(f) \leq 2G\,\mathcal{R}_n(\mathcal{F}_{\phi,L}^\circ) \leq \frac{4GLc^2}{\sqrt{n}}. \tag{12}$$

In particular, this $O(n^{-1/2})$ rate in the number of samples is the same as the one obtained for standard kernel models (see [30]). Hence, despite the proposed model is expressed with respect to a matrix (and so in principle a squared number of degrees of freedom), complementing from a statistical viewpoint the results obtained in Thm. 2 and Remark 1 from a representation viewpoint.

## 5 Extensions: Integral Constraints and Output in Convex Cones

In this section we cover two extensions. The first one generalizes the optimization problem in Eq. (5) to include linear constraints on the integral of the model, in order to deal with problems like density estimation in Example 1. The second formalizes models with outputs in convex cones, which is crucial when dealing with problems like multivariate quantile estimation [10], detailed in Sec. 6.

**Constraints on the integral and other linear constraints.** We can extend the definition of the problem in Eq. (5) to take into account constraints on the integral of the model. Indeed by linearity of integration and trace, we have the following (proof in Appendix B.10).

**Proposition 4** (Integrability in closed form, **P4**). *Let $A \in \mathcal{S}(\mathcal{H})$ with $A$ bounded and $\phi$ uniformly bounded. Let $p : \mathcal{X} \to \mathbb{R}$ be an integrable function. There exists a trace class operator $W_p \in \mathcal{S}(\mathcal{H})$ such that $\int_\mathcal{X} f_A(x)p(x)dx = \mathrm{Tr}(AW_p)$ and $W_p = \int_\mathcal{X} \phi(x)\phi(x)^\top p(x)dx$.*

The result can be extended to derivatives and more general linear functionals on $f_A$ (see Appendix B.10). In particular, note that if we consider the *empirical feature map* $\Phi$ in Eq. (8), which characterizes the optimal solution of Eq. (5), by Thm. 1, we have that $W_p$ is defined explicitly as $W_p = \mathbf{V}^{-\top}\mathbf{M}_p\mathbf{V}^{-1}$ with $(\mathbf{M}_p)_{i,j} = \int k(x, x_i)k(x, x_j)p(x)dx$, for $i, j = 1, \dots, n$ and it is computable in closed form. Then, assuming an equality and an inequality constraint on the integral w.r.t. two functions $p$ and $q$ and two values $c_1, c_2 \in \mathbb{R}$, the resulting problem takes the following finite-dimensional form

$$\min_{\mathbf{A} \in \mathcal{S}(\mathbb{R}^n)} \quad L(\tilde{f}_\mathbf{A}(x_1), \dots, \tilde{f}_\mathbf{A}(x_n)) + \Omega(\mathbf{A}), \tag{13}$$
$$\text{s.t.} \quad \mathbf{A} \succeq 0, \ \mathrm{Tr}(\mathbf{A}W_p) = c_1, \ \mathrm{Tr}(\mathbf{A}W_q) \leq c_2.$$

**Representing function with outputs in convex polyhedral cones.** We represent a vector-valued function with our model as the juxtaposition of $p$ scalar valued models, with $p \in \mathbb{N}$, as follows

$$f_{A_1 \cdots A_p}(x) = (f_{A_1}(x), \dots, f_{A_p}(x)) \in \mathbb{R}^p, \qquad \forall\, x \in \mathcal{X}.$$

We recall that a convex polyhedral cone $\mathcal{Y}$ is defined by a set of inequalities as follows

$$\mathcal{Y} = \{y \in \mathbb{R}^p \mid c^{1\top}y \geq 0, \dots, c^{h\top}y \geq 0\}, \tag{14}$$

for some $c^1, \dots, c^h \in \mathbb{R}^p$ and $h \in \mathbb{N}$. Let us now focus on a single constraint $c^\top y \geq 0$. Note that, by definition of positive operator (i.e., $A \succeq 0$ implies $v^\top A v \geq 0$ for any $A$), we have that $\sum_{s=1}^p c_s A_s \succeq 0$ implies $\phi(x)^\top (\sum_{s=1}^p c_s A_s)\phi(x) \geq 0$ for any $x \in \mathcal{X}$, which, by linearity of the inner product and the definition of $f_{A_1 \cdots A_p}$ is equivalent to $c^\top f_{A_1 \cdots A_p}(x) \geq 0$. From this reasoning we derive the following proposition (see complete proof in Appendix B.11).

**Proposition 5.** *Let $\mathcal{Y}$ be defined as in Eq. (14). Let $A_1, \dots, A_p \in \mathcal{S}(\mathcal{H})$. Then the following holds*

$$\sum_{s=1}^p c_s^t A_s \succeq 0 \quad \forall t = 1, \dots, h \qquad \Rightarrow \qquad f_{A_1 \cdots A_p}(x) \in \mathcal{Y} \quad \forall x \in \mathcal{X}.$$

Note that the set of constraints on the l.h.s. of the equation above defines in turn a convex set on $A_1, \ldots, A_p$. This means that we can use it to constrain a convex optimization problem over the space of the proposed vector-valued models as follows

$$\min_{A_1, \ldots, A_p \in \mathcal{S}(\mathcal{H})} L(f_{A_1 \cdots A_p}(x_1), \ldots, f_{A_1 \cdots A_p}(x_n)) + \sum_{s=1}^{p} \Omega(A_s) \qquad (15)$$

$$\text{s.t.} \quad \sum_{s=1}^{p} c_s^t A_s \succeq 0, \quad \forall \, t = 1, \ldots, h.$$

By Proposition 5, the function $f_{A_1^* \cdots A_p^*}$, where $(A_1^*, \ldots, A_p^*)$ is the minimizer above, will be a function with output in $\mathcal{Y}$. Moreover, the formulation above admits a finite-dimensional representation analogous to the one for non-negative functions, as stated below (see proof in Appendix B.12)

**Theorem 6** (Representer theorem for model with output in convex polyhedral cones). *Under the assumptions of Thm. 1, the problem in Eq. (15) admits a minimizer $(A_1^*, \cdots, A_p^*)$ of the form*

$$A_s^* = \sum_{i,j=1}^{n} [\mathbf{B}_s]_{i,j} \phi(x_i)\phi(x_j)^\top \implies (f_*(x))_s = \sum_{i,j=1}^{n} [\mathbf{B}_s]_{i,j} k(x_i, x)k(x_j, x), \quad s = 1, \ldots, p,$$

*where $f_* := f_{(A_1^*, \ldots, A_p^*)}$ is the corresponding function and the $\mathbf{B}_s \in \mathcal{S}(\mathbb{R}^n)$ are symmetric $n \times n$ matrices which satisfy the conic constraints $\sum_{s=1}^{p} c_s^t \mathbf{B}_s \succeq 0, \ t = 1, \ldots, h.$*

**Remark 2** (Efficient representations when the ambient space of $\mathcal{Y}$ is high-dimensional). *When $p \gg h$, or when $\mathcal{Y}$ is a polyhedral cone with $\mathcal{Y} \subset \mathcal{G}$ and $\mathcal{G}$ an infinite-dimensional space, it is still possible to have an efficient representation of functions with output in $\mathcal{Y}$ by using the representation of $\mathcal{Y}$ in terms of* conical hull *[12], i.e., $\mathcal{Y} = \{\sum_{i=1}^{t} \alpha_i y_i \mid \alpha_i \geq 0\}$ for some $y_1, \ldots, y_t$ and $t \in \mathbb{N}$. In particular, given $A_1, \ldots, A_t \succeq 0$, the model $f_{A_1 \ldots A_t}(x) = \sum_{i=1}^{t} f_{A_i}(x) y_i$ satisfies $f_{A_1 \ldots A_t}(x) \in \mathcal{Y}$ for any $x \in \mathcal{X}$. Moreover it is possible to derive a representer theorem as Thm. 6.*

**Remark 3.** *By extending this approach, we believe it is possible to model (a) functions with output in the cone of positive semidefinite matrices, (b) convex functions. We leave this for future work.*

# 6  Numerical Simulations

In this section, we provide illustrative experiments on the problems of density estimation, regression with Gaussian heteroscedastic errors, and multiple quantile regression. We derive the algorithm according to the finite-dimensional formulation in Eq. (13) for non-negative functions with constraints on the integral, and to Eq. (15) with the finite-dimensional representation suggested by Thm. 6. Optimization is performed applying FISTA [6] on the dual of the resulting formulations. More details on implementation and the specific formulations are given below and in Appendix E. The algorithms are compared with careful implementations of Eq. (1) with the models presented in Sec. 2.1, i.e., partially non-negative models (PNM), non-negative coefficients models (NCM) and generalized linear models (GLM). For all methods we used $\Omega(A) = \lambda_1 \|A\|_* + \frac{\lambda_2}{2} \|A\|_F^2$ or $\Omega(w) = \frac{\lambda}{2} \|w\|^2$. We used the Gaussian kernel $k(x, x') = \exp(-\|x - x'\|^2/(2\sigma^2))$ with width $\sigma$. Full cross-validation has been applied to each model independently, to find the best $\lambda, \lambda_1, \lambda_2$ (see Appendix E). The code for these experiments is available on GitHub (`https://github.com/umarteau/non_negative_model`).

**Density estimation.** This problem is illustrated in Example 1. Here we considered the *log-likelihood loss* as a measure of error, i.e., $L(z_1, \ldots, z_n) = -\frac{1}{n} \sum_{i=1}^{n} \log(z_i)$, which is jointly convex and with an efficient proximal operator [13]. We recall that the problems are constrained to output a function whose integral is 1. In Fig. 1, we show the experiment on $n = 50$ i.i.d. points sampled from $\rho(x) = \frac{1}{2}\mathcal{N}(-1, 0.3) + \frac{1}{2}\mathcal{N}(1, 0.3)$ and where for all the models we used $\sigma = 1$, to illustrate pictorially the main interesting behaviors. Note that PNM *(left)* is non-negative on the training points, but it achieves negative values on the regions not covered by examples. This effect is worsened by the constraint on the integral that borrows areas from negative regions to reduce the log-likelihood on the dataset. NCM *(center-left)* produces a function whose integral is one and that is non-negative everywhere, but the poor approximation properties of the model do not allow to fit the density of interest (see Example 2). GLM *(center-right)* produces a function that is non-negative and approximates quite well $\rho$, however, the obtained function does not sum to one, but to 0.987, since the integral constraint can be enforced only approximately via Monte Carlo sampling (GLM does

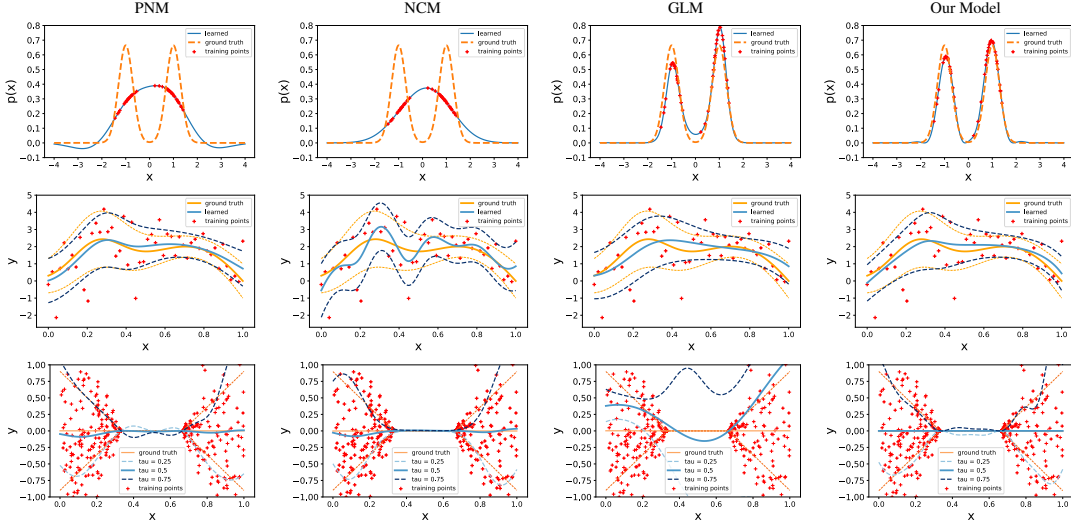

Figure 1: Details in Sec. 6. *(top)* density estimation, *(center)* regression with Gaussian heteroscedastic errors. *(bottom* multiple quantile regression. Shades of blue: estimated curves. Orange: ground truth. Models: *(left)* PNM, *(center-left)* NCM, *(center-right)*, GLM, *(right)* Our model.

not satisfy **P4**). Estimating the integral is easy in low dimensions but becomes soon impractical in higher dimensions [26]. Finally the proposed model *(right)* leads to a convex problem and produces a non-negative function whose integral is 1 and that fits the density $\rho$ quite well. In Appendix E we obtain a similar result for a multivariate version of this experiment with $d = 10$ and $n = 1000$, where we cross-validated $\sigma$ for each algorithm.

**Heteroscedastic Gaussian process estimation.** The goal is to estimate $\mu : \mathbb{R} \to \mathbb{R}$ and $v : \mathbb{R} \to \mathbb{R}_+$ determining the conditional density $\rho$ of the form $\rho(y|x) = (2\pi v(x))^{-1/2} \exp(-(y - \mu(x))^2/(2v(x)))$ from which the data are sampled. The considered functional corresponds to the negative log-likelihood, i.e., $L = \sum_{i=1}^{n} \frac{1}{2} \log v(x_i) + (y_i - \mu(x_i))^2/(2v(x_i))$ that becomes convex in $\eta, \theta$ via the so called *natural parametrization* $\eta(x) = \mu(x)/v(x)$ and $\theta(x) = 1/v(x)$ [21]. We used a linear model to parametrize $\eta$ and the non-negative models for $\theta$. The experiment on the same model of [21, 38] is reported in Fig. 1. Modeling $\theta$ via PNM *(left)* leads to a convex problem and reasonable performance. In particular, the fact that $\theta = 0$ corresponds to $v = +\infty$ prevents the model for $\theta$ from crossing zero. NCM *(center-left)* leads to a convex problem, but very sensitive to the kernel width $\sigma$ and with poor approximation properties. GLM *(center-right)* leads to a non-convex problem and we need to restart the method randomly to have a reasonable convergence. Our model *(right)* leads to a convex problem and produces a non-negative function for $\theta$, that fits well the observed data.

**Multiple quantile regression.** The goal here is to estimate multiple quantiles of a given conditional distribution $P(Y|x)$. Given $\tau \in (0, 1)$, $q_\tau$ defined by $P(Y > q_\tau(x)|x) = \tau$ is the $\tau$-quantile of $\rho$. By construction $0 < \tau_{-h} \le \cdots \le \tau_h < 1$ implies $q_{\tau_{-h}}(x) \le \cdots \le q_{\tau_h}(x)$. If we denote by $\mathbf{q} : \mathcal{X} \to \mathbb{R}^{2h+1}$ the list of quantiles, we have by construction $\mathbf{q}(x) \in \mathcal{Y}$ where $\mathcal{Y}$ is a convex cone $\mathcal{Y} = \{y \in \mathbb{R}^h \mid y_{-h} \le \cdots \le y_h\}$. To regress quantiles, we used the pinball loss $L_\tau$ (convex, non-smooth) considered in [20, 32], obtaining $L = \sum_{j=-h}^{h} \sum_{i=1}^{n} L_{\tau_j}(f(x_i), y_i)$. In Fig. 1, we used $\tau_{-1} = \frac{1}{4}, \tau_0 = \frac{1}{2}, \tau_1 = \frac{3}{4}$. Using PNM, *(left)* the ordering is enforced by explicit constraints on the observed dataset [33, 10]. The resulting problem is convex. However, in regions with low density of points, PNM quantiles do not respect their natural order. To enforce the order constraint, a fine grid covering the space would be needed as in [33]. For NCM, GLM and our model, we represented the quantiles as $q_{\tau_{\pm j}} = q_{\tau_0} \pm \sum_{i=1} v_{\pm i}$ where the $v$'s are non-negative functions and $q_{\tau_0}$, with $\tau_0 = \frac{1}{2}$, is the median and is modeled by a linear model. NCM *(center-left)* leads to a convex problem and quantiles that respect the ordering, but the estimation is very sensitive to the chosen $\sigma$ and has poor approximation properties. GLM *(center-right)* leads to a non-convex non-differentiable problem, with many local minima, which is difficult to optimize with standard techniques (see Appendix E). GLM does not succeed in approximating the quantiles. Our model *(right)* leads to a convex optimization problem that approximates the quantiles relatively well and preserves their natural order everywhere.

## Broader Impact

This work does not present any foreseeable societal consequence.

## Acknowledgments and Disclosure of Funding

This work was funded in part by the French government under management of Agence Nationale de la Recherche as part of the "Investissements d'avenir" program, reference ANR-19- P3IA-0001 (PRAIRIE 3IA Institute). We also acknowledge support of the European Research Council (grant SEQUOIA 724063).

## Footnotes

[1]Note that the model in Eq. (4) has already been considered in [4] with a similar goal as ours. However, this workshop publication has only be lightly peer-reviewed, the representer theorem they propose is incorrect,

[2]Given the eigendecomposition $\mathbf{A} = \mathbf{U}\mathbf{\Lambda}\mathbf{U}^\top$ with $\mathbf{U} \in \mathbb{R}^{n \times n}$ unitary and $\mathbf{\Lambda} \in \mathbb{R}^{n \times n}$ diagonal, then $[\mathbf{A}]_- = \mathbf{U}\mathbf{\Lambda}_-\mathbf{U}^\top$, with $\mathbf{\Lambda}_-$ diagonal, defined as $(\mathbf{\Lambda}_-)_{i,i} = \min(0, \mathbf{\Lambda}_{i,i})$ for $i = 1, \dots, n$.

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
