[Supplementary Material]

# A  Notation and basic definitions

- $\mathcal{H}$ is a separable Hilbert space.
- $\mathcal{X}$ is a Polish space (we will require explicitly compactness in some theorems).
- $\phi : \mathcal{X} \to \mathcal{H}$ is a continous map. We also assume it to be uniformly bounded i.e. $\sup_{x \in \mathcal{X}} \|\phi(x)\| \le c$ for some $c \in (0, \infty)$, if not differently stated.
- $k(x, x') := \phi(x)^\top \phi(x')$ is the *kernel* function associated to the feature map $\phi$ [SS02, BTA11].

# B  Proofs and additional discussions

## B.1  Proof of Proposition 1

In this section, let us extend the definition in Eq. (4) to any operator $A \in \mathcal{S}(\mathcal{H})$, without the implied positivity restriction (in Eq. (4), we ask that $A \succeq 0$) :

$$\forall A \in \mathcal{S}(\mathcal{H}), \ \forall x \in \mathcal{X}, \ f_A(x) := \phi(x)^\top A \phi(x). \tag{4bis}$$

*Proof of Proposition 1.* To prove linearity, let $A, B \in \mathcal{S}(\mathcal{H})$ and $\alpha, \beta \in \mathbb{R}$. Since $\mathcal{S}(\mathcal{H})$ is a vector space, $\alpha A + \beta B \in \mathcal{S}(\mathcal{H})$. Let $x \in \mathcal{X}$. By definition for the first equality and linearity for the second,

$$f_{\alpha A + \beta B}(x) = \phi(x)^\top (\alpha A + \beta B) \phi(x) = \alpha \phi(x)^\top A \phi(x) + \beta \phi(x)^\top B \phi(x).$$

Finally, since by definition, $f_A(x) = \phi(x)^\top A \phi(x)$ and $f_B(x) = \phi(x)^\top B \phi(x)$, it holds :

$$f_{\alpha A + \beta B}(x) = \alpha \phi(x)^\top A \phi(x) + \beta \phi(x)^\top B \phi(x) = \alpha f_A(x) + \beta f_B(x).$$

Since this holds for all $x \in \mathcal{X}$, this shows $f_{\alpha A + \beta B} = \alpha f_A + \beta f_B$.

To prove the non-negativity, assume now that $A \succeq 0$. By definition of of positive semi-definiteness,

$$\forall h \in \mathcal{H}, \ h^\top A h \ge 0.$$

In particular, for any $x \in \mathcal{X}$, the previous inequality applied to $h = \phi(x)$ yields

$$f_A(x) = \phi(x)^\top A \phi(x) \ge 0.$$

Hence, $f_A \ge 0$.

$\square$

## B.2  Proof of Proposition 2

Recall the definition of $f_A$ for any $A \in \mathcal{S}(\mathcal{H})$ in Eq. (4bis). We have the lemma:

**Lemma 1** (Linearity of evaluations)**.** *Let $x_1, \ldots, x_n \in \mathcal{X}$. Then the map*

$$A \in \mathcal{S}(\mathcal{H}) \mapsto (f_A(x_i))_{1 \le i \le n} \in \mathbb{R}^n$$

*is linear from $\mathcal{S}(\mathcal{H})$ to $\mathbb{R}^n$.*

*Proof.* This just follows from the fact that the definition of $f_A(x_i)$, $f_A(x_i) := \phi(x_i)^\top A \phi(x_i)$, is linear in $A$. $\square$

**Proof of Proposition 2.** Let $L : \mathbb{R}^n \to \mathbb{R}$ be a jointly convex function and $x_1, \ldots, x_n \in \mathcal{X}$. The function $A \in \mathcal{S}(\mathcal{H}) \mapsto L(f_A(x_1), \ldots, f_A(x_n))$ can be written $L \circ R$, where

$$R : A \in \mathcal{S}(\mathcal{H}) \mapsto (f_A(x_i))_{1 \le i \le n} \in \mathbb{R}^n.$$

Since $L$ is convex, and $R$ is linear by Lemma 1, their composition is convex.
Moreover, since $\mathcal{S}(\mathcal{H})_+$ is a convex subset of $\mathcal{S}(\mathcal{H})$, the restriction of $A \in \mathcal{S}(\mathcal{H}) \mapsto L(f_A(x_1), \ldots, f_A(x_n))$ on $\mathcal{S}(\mathcal{H})_+$ is also convex. $\square$

## B.3 Proof of Thm. 1

In this section, we prove Thm. 1 for a more general class of spectral regularizers.

### B.3.1 Compact operators and spectral functions

In this section, we briefly introduce compact self-adjoint operators and the spectral theory of compact self-adjoint operators. For more details, see for instance [GGK04]. We start by defining a compact self-adjoint operator (see Section2.16 of [GGK04]) and stating its main properties:

**Definition 1** (compact operators). *Let $\mathcal{H}$ be a separable Hilbert space. A bounded self-adjoint operator $A \in \mathcal{S}(\mathcal{H})$ is said to be compact if its range is included in a compact set. We denote with $\mathcal{S}_\infty(\mathcal{H})$ the set of compact self adjoint operators on $\mathcal{H}$. It is a closed subspace of $\mathcal{S}(\mathcal{H})$ for the operator norm and the closure of the set of finite rank operators.*

**Proposition 6** (Spectral theorem [GGK04]). *Let $\mathcal{H}$ be a separable Hilbert space and let $A$ be a compact self adjoint operator on $\mathcal{H}$. Then there exists a spectral decomposition of A, i.e., an orthonormal system $(u_k) \in \mathcal{H}$ of eigenvectors of A and corresponding eigenvalues $(\sigma_k)$ such that for all $h \in \mathcal{H}$, it holds*

$$Ah = \sum_k \sigma_k u_k^\top h \, u_k =: \left( \sum_k \sigma_k u_k u_k^\top \right) h.$$

*Moreover, if $\sigma_k$ is an infinite sequence, it converges to zero.*
*Furthermore, we say that the orthonormal system $(u_k)$ of eigenvectors of A and the corresponding eigenvalues $(\sigma_k)$ is a* basic system *of eigenvectors of A if all the $\sigma_k$ are non zero. In this case, if $P_0$ denotes the orthogonal projection on $\mathrm{Ker}(A)$, then it holds*

$$\forall h \in \mathcal{H}, \ h = \Pi_0 \, h + \sum_k u_k u_k^\top \, h$$

In what follows, to simplify notations, we will usually write $A = U \, \mathrm{Diag}(\sigma) U^\top$ in order to denote a basic system of eigenvectors of $A$. Moreover, if $A$ is positive semi-definite, we will assume that the eigenvalues are sorted in decreasing order, i.e., $\sigma_{k+1} \leq \sigma_k$.

**Definition 2** (Spectral function on $\mathcal{S}_\infty(\mathcal{H})$ [GGK04]). *Let $q : \mathbb{R} \to \mathbb{R}$ be a lower semi-continuous function such that $q(0) = 0$. Let $\mathcal{H}$ be any separable Hilbert space. For any $A \in \mathcal{S}_\infty(\mathcal{H})$ and any basic system $A = U \, \mathrm{Diag}(\sigma) U^\top$, we define the spectral function $q$*

$$q(A) = U \, \mathrm{Diag}(q(\sigma))) U^T = \sum_k q(\sigma_k) u_k u_k^\top.$$

### B.3.2 Classes of regularizers

Let us now state our main assumption on regularizers.

**Assumption 1** (Assumption on regularizers). *$\Omega$ is of of the form*

$$\forall A \in \mathcal{S}(\mathcal{H}), \ \Omega(A) = \begin{cases} \mathrm{Tr}(q(A)) = \sum_k q(\sigma_k) & \text{if } A = U \, \mathrm{Diag}(\sigma) U^\top \in \mathcal{S}_\infty(\mathcal{H}), \ \sum_k q(\sigma_k) < \infty \\ +\infty & \text{otherwise,} \end{cases}$$

*where $q : \mathbb{R} \to \mathbb{R}_+$ is:*

- *non-decreasing on $\mathbb{R}_+$ with $q(0) = 0$;*

- *lower semi-continuous;*

- *$q(\sigma) \underset{|\sigma| \to +\infty}{\longrightarrow} +\infty$.*

Note that in this case, $\Omega$ is defined on $\mathcal{S}(\mathcal{H})$ for any Hilbert space $\mathcal{H}$.

**Remark 4.** $\Omega(A) = \lambda_1 \|A\|_\star + \frac{\lambda_2}{2} \|A\|_F^2$ *satisfies Assumption 1, with $q(\sigma) = \lambda_1 |\sigma| + \lambda_2 \, \sigma^2$.*

**Lemma 2** (Properties of $\Omega$). *Let $\Omega$ satisfying Assumption 1. Then the following properties hold.*

(i) *For any separable Hilbert spaces $\mathcal{H}_1, \mathcal{H}_2$ and any linear isometry $O : \mathcal{H}_1 \to \mathcal{H}_2$, i.e., such that $O^*O = I_{\mathcal{H}_1}$, it holds*

$$\forall A \in \mathcal{S}(\mathcal{H}_1), \ \Omega(OAO^*) = \Omega(A).$$

(ii) *For any separable Hilbert space $\mathcal{H}$ and any orthogonal projection $\Pi \in \mathcal{S}(\mathcal{H}_1)$, i.e., satisfying $\Pi = \Pi^*$, $\Pi^2 = \Pi$, it holds*

$$\forall A \succeq 0, \ \Omega(\Pi A \Pi) \le \Omega(A).$$

(iii) *For any finite dimensional Hilbert space $\mathcal{H}_n$,*

$$\Omega \text{ is lower semi-continuous (l.s.c)}, \qquad \Omega(A) \underset{\|A\|_{op} \to +\infty}{\longrightarrow} +\infty$$

*where we denoted by $\| \cdot \|_{op}$ the operator norm.*

*Proof.*     (i) Write $A = \sum_k \sigma_k u_k u_k^\top$ where the $(u_k)$ form a basic system of eigen-vectors for $A$. The $(v_k) = (Ou_k)$ form a basic system of eigen-vectors for $OAO^*$, as

$$OAO^* = \sum_k \sigma_k v_k v_k^\top, \qquad \sigma_k \ne 0.$$

Hence, by definition, $q(OAO^*) = \sum_k q(\sigma_k) v_k v_k^\top$. By definition of the trace, we have

$$\Omega(OAO^*) = \sum_k q(\sigma_k) = \Omega(A).$$

(ii) Let $A$ be a compact self-adjoint semi-definite operator. Let $A = U \operatorname{Diag}(\sigma) U^\top$ be a basic system of eigenvectors of $A$, where the $\sigma_k$ are positive and in decreasing order. Define $B = U \operatorname{Diag}(\sqrt{\sigma}) U^\top$ and note that in this case, $A = B^2 = B^*B$. Using Exercise 23 of [GGK04], we have that for any orthogonal projection operator $\Pi$ and any index $k$, $\sigma_k(\Pi B^* B \Pi) \le \sigma_k(B^*B)$ and hence $\sigma_k(\Pi A \Pi) \le \sigma_k(A)$. Since $q$ is non decreasing, it holds $q(\sigma_k(\Pi A \Pi)) \le q(\sigma_k(A))$ and hence

$$\Omega(\Pi A \Pi) = \sum_k q(\sigma_k(\Pi A \Pi)) \le \sum_k q(\sigma_k(A)) = \Omega(A).$$

(iii) Let $\mathcal{H}_n$ be a finite dimensional Hilbert space and let $\| \cdot \|_{op}$ be the operator norm on $\mathcal{S}(\mathcal{H}_n)$. If $q$ is continuous, then $A \in \mathcal{H}_n \mapsto q(A)$ is continuous and hence $\Omega$ is continuous (since the trace is continuous in finite dimensions). Now assume $q$ is lower semi-continuous, and define for $n \in \mathbb{N}$, $q_n(t) := \inf_{s \in \mathbb{R}} q(s) + n|t - s|$. We have $q_n \ge 0$, $q_n(0) = 0$ $q_n$ is uniformly continuous and $q_n$ is an increasing sequence of functions such that $q_n \to q$ point-wise. Now it is easy to see that $\operatorname{Tr}(q(A)) = \sup_n \operatorname{Tr}(q_n(A))$ and hence $\Omega$ is lower semi-continuous as a supremum of continuous functions.
The fact that $\Omega$ goes to infinity is a direct consequence of the fact that $q$ goes to infinity, by Assumption 1.

$\square$

**Remark 5.** *The three conditions of the previous lemma are in fact the only conditions needed in the proof. We could loosen Assumption 1 to satisfy only these three properties.*

### B.3.3   Finite-dimensional representation and existence of a solution

Fix $n \in \mathbb{N}$, a loss function $L : \mathbb{R}^n \to \mathbb{R} \cup \{+\infty\}$, a separable Hilbert space $\mathcal{H}$, a regularizer $\Omega$ on $\mathcal{S}(\mathcal{H})$ a feature map $\phi : \mathcal{X} \to \mathcal{H}$ and points $(x_1, ..., x_n) \in \mathcal{X}^n$.

Recall the problem in Eq. (5):

$$\inf_{A \succeq 0} L(f_A(x_1), \dots, f_A(x_n)) + \Omega(A). \tag{5}$$

Define $\mathcal{H}_n$ to be the finite-dimensional subset of $\mathcal{H}$ spanned by the $\phi(x_i)$, i.e.,

$$\mathcal{H}_n := \mathrm{span}\,(\phi(x_i))_{1 \le i \le n} = \left\{ \sum_{i=1}^{n} \alpha_i \phi(x_i) \; : \; \alpha \in \mathbb{R}^n \right\}.$$

Define $\Pi_n$ is the orthogonal projection on $\mathcal{H}_n$, i.e.,

$$\Pi_n \in \mathcal{S}(\mathcal{H}), \; \Pi_n^2 = \Pi_n, \; \mathrm{range}(\Pi_n) = \mathcal{H}_n.$$

Define $\mathcal{S}_n(\mathcal{H})_+$ to be the following subspace of $\mathcal{S}(\mathcal{H})_+$ :

$$\mathcal{S}_n(\mathcal{H})_+ := \Pi_n \mathcal{S}(\mathcal{H})_+ \Pi_n = \{ \Pi_n A \Pi_n \; : \; A \in \mathcal{S}(\mathcal{H})_+ \}.$$

**Proposition 7.** *Let $L$ be a lower semi-continuous function which is bounded below, and assume $\Omega$ satisfies Assumption 1. Then Eq. (5) has a solution $A^*$ which is in $\mathcal{S}_n(\mathcal{H})_+$.*

*Proof.* In this proof, denote by $J$ the function defined by

$$\forall A \in \mathcal{S}(\mathcal{H}), \; J(A) := L(f_A(x_1), ..., f_A(x_n)) + \Omega(A).$$

Our goal is to prove that the problem $\inf_{A \in \mathcal{S}(\mathcal{H})_+} J(A)$ has a solution which is in $\mathcal{S}_n(\mathcal{H})_+$, i.e., of the form $\Pi_n A \Pi_n$ for some $A \in \mathcal{S}(\mathcal{H})_+$.

**1.** Let us start by fixing $A \in \mathcal{S}(\mathcal{H})_+$.
First note that since $\Pi_n$ is the orthogonal projection on $\mathrm{span}(\phi(x_i))_{1 \le i \le n}$, in particular $\Pi_n \phi(x_i) = \phi(x_i)$ for all $1 \le i \le n$. Thus, for any $1 \le i \le n$,

$$f_A(x_i) = \phi(x_i)^\top A \phi(x_i) = \phi(x_i)^\top \Pi_n A \Pi_n \phi(x_i) = f_{\Pi_n A \Pi_n}(x_i).$$

Here, the first and last equalities come from the definition of $f_A$ and $f_{\Pi_n A \Pi_n}$. Thus,

$$J(A) = L(f_{\Pi_n A \Pi_n}(x_1), ..., f_{\Pi_n A \Pi_n}(x_n)) + \Omega(A).$$

Now since $\Omega$ satisfies Assumption 1, by the second point of Lemma 2, it holds $\Omega(\Pi_n A \Pi_n) \le \Omega(A)$, hence

$$J(\Pi_n A \Pi_n) \le J(A).$$

This last inequality combined with the fact that $\mathcal{S}_n(\mathcal{H})_+ = \Pi_n \mathcal{S}(\mathcal{H})_+ \Pi_n \subset \mathcal{S}(\mathcal{H})_+$ show that

$$\inf_{A \in \mathcal{S}_n(\mathcal{H})_+} J(A) = \inf_{A \succeq 0} J(A). \qquad (16)$$

**2.** Let us now show that $\inf_{A \in \mathcal{S}_n(\mathcal{H})_+} J(A)$ has a solution. Let us exclude the case where $J = +\infty$, in which case $A = 0$ can be taken to be a solution.

Let $V_n$ be the injection $V_n : \mathcal{H}_n \hookrightarrow \mathcal{H}$. Note that $V_n V_n^* = \Pi_n$ and $V_n^* V_n = I_{\mathcal{H}_n}$. These simple facts easily show that

$$\mathcal{S}_n(\mathcal{H})_+ = V_n \mathcal{S}(\mathcal{H}_n)_+ V_n^* = \left\{ V_n \tilde{A} V_n^* \; : \; \tilde{A} \in \mathcal{S}(\mathcal{H}_n)_+ \right\}.$$

Thus, our goal is to show that $\inf_{\tilde{A} \in \mathcal{S}(\mathcal{H}_n)_+} J(V_n A V_n^*)$ has a solution.

By the first point of Lemma 2, since $V_n^* V_n = I_{\mathcal{H}_n}$, it holds

$$\forall \tilde{A} \in \mathcal{S}(\mathcal{H}_n), \; \Omega(V_n \tilde{A} V_n^*) = \Omega(\tilde{A}) \implies J(V_n \tilde{A} V_n^*) = L(f_{V_n \tilde{A} V_n^*}(x_1), ..., f_{V_n \tilde{A} V_n^*}(x_n)) + \Omega(\tilde{A}).$$

Let $\tilde{A}_0 \in \mathcal{S}(\mathcal{H}_n)_+$ be a point such that $J_0 := J(V_n \tilde{A}_0 V_n^*) < \infty$. Let $c_0$ be a lower bound for $L$. By the third point of Lemma 2, there exists a radius $R_0$ such that for all $\tilde{A} \in \mathcal{S}(\mathcal{H}_n)$,

$$\|\tilde{A}\|_F > R_0 \implies \Omega(\tilde{A}) > J_0 - c_0.$$

Since $c_0$ is a lower bound for $L$, this implies

$$\inf_{\tilde{A} \in \mathcal{S}(\mathcal{H}_n)_+} J(V_n \tilde{A} V_n^*) = \inf_{\tilde{A} \in \mathcal{S}(\mathcal{H}_n)_+, \, \|\tilde{A}\|_F \leq R_0} J(V_n \tilde{A} V_n^*).$$

Now since $L$ is lower semi-continuous, $\Omega$ is lower semi-continuous by the last point of Lemma 2, and $\tilde{A} \mapsto (f_{V_n \tilde{A} V_n^*}(x_i))_{1 \leq i \leq n}$ is linear hence continuous, the mapping $A \mapsto J(V_n \tilde{A} V_n^*)$ is lower semi-continuous. Hence, it reaches its minimum on any non empty compact set. Since $\mathcal{H}_n$ is finite dimensional, the set $\left\{ \tilde{A} \in \mathcal{S}(\mathcal{H}_n)_+ \; : \; \|\tilde{A}\|_F \leq R_0 \right\}$ is compact (closed and bounded) and non empty since it contains $\tilde{A}_0$, and hence there exists $\tilde{A}_* \in \mathcal{S}(\mathcal{H}_n)_+$ such that $J(V_n \tilde{A}_* V_n^*) = \inf_{\tilde{A} \in \mathcal{S}(\mathcal{H}_n)_+, \, \|\tilde{A}\|_F \leq R_0} J(V_n \tilde{A} V_n^*)$. Going back up the previous equalities, this shows that $A_* = V_n \tilde{A}_* V_n^* \in \mathcal{S}_n(\mathcal{H})_+$ and $J(A_*) = \inf_{A \succeq 0} J(A)$. □

### B.3.4 Proof of Thm. 1

We will prove the following Thm. 7 whose statement is that of Thm. 1 with more general assumptions.

**Theorem 7.** *Let $L$ be lower semi-continuous and bounded below, and $\Omega$ satisfying Assumption 1. Then Eq. (5) has a solution $A_*$ which can be written in the form*

$$\sum_{i,j=1}^{n} \mathbf{B}_{ij} \phi(x_i) \phi(x_j)^\top, \qquad \text{for some matrix } \mathbf{B} \in \mathbb{R}^{n \times n}, \; \mathbf{B} \succeq 0.$$

*Moreover, if $L$ is convex, and $\Omega$ is of the form Eq. (6) with $\lambda_2 > 0$, this solution is unique. By Eq. (4), $A_*$ corresponds to a function of the form*

$$f_*(x) = \sum_{i,j=1}^{n} \mathbf{B}_{ij} k(x, x_i) k(x, x_j).$$

**Lemma 3.** *The set $\mathcal{S}_n(\mathcal{H})_+$ can be represented in the following way*

$$\mathcal{S}_n(\mathcal{H})_+ = \left\{ \sum_{1 \leq i,j \leq n} \mathbf{B}_{i,j} \phi(x_i) \phi(x_j)^\top, \; : \; \mathbf{B} \in \mathbb{R}^{n \times n}, \; \mathbf{B} \succeq 0 \right\}.$$

*In particular, for any $A \in \mathcal{S}_n(\mathcal{H})_+$, there exists a matrix $\mathbf{B} \in \mathbb{R}^{n \times n}$, $\mathbf{B} \succeq 0$ such that*

$$A = \sum_{1 \leq i,j \leq n} \mathbf{B}_{i,j} \phi(x_i) \phi(x_j)^\top \implies \forall x \in \mathcal{X}, \; f_A(x) = \sum_{1 \leq i,j \leq n} \mathbf{B}_{i,j} k(x_i, x) k(x_j, x).$$

*Proof.* Define $S_n : \mathcal{H} \to \mathbb{R}^n$ to be the operator such that

$$\forall h, \; S_n(h) = \left( h^\top \phi(x_i) \right)_{1 \leq i \leq n},$$

with adjoint $S_n^* : \mathbb{R}^n \to \mathcal{H}$ such that

$$\forall \alpha \in \mathbb{R}^n, \; S_n^* \alpha = \sum_{i=1}^{n} \alpha_i \phi(x_i).$$

Note that for any $\mathbf{B} \in \mathbb{R}^{n \times n}$, $S_n^* \mathbf{B} S_n = \sum_{i,j} \mathbf{B}_{i,j} \phi(x_i) \phi(x_j)^\top$.

**1. Proving $\mathcal{S}_n(\mathcal{H})_+ \subset \left\{ \sum_{1 \leq i,j \leq n} \mathbf{B}_{i,j} \phi(x_i) \phi(x_j)^\top, \; : \; \mathbf{B} \in \mathbb{R}^{n \times n}, \; \mathbf{B} \succeq 0 \right\}$.** Let $\Pi_n A \Pi_n \in \mathcal{S}_n(\mathcal{H})_+$. Using the previous equality, we want to show there exists $\mathbf{B} \in \mathbb{R}^{n \times n}$, $\mathbf{B} \succeq 0$ such that $\Pi_n A \Pi_n = S_n^* \mathbf{B} S_n$. Using Lemma 4, we see that $\Pi_n$ can be written in the form $S_n^* T_n$ where $T_n : \mathcal{H} \to \mathbb{R}^n$ (write $\Pi_n = O_n O_n^*$ and note that $O_n$ is of the form $S_n^* \tilde{O}_n$). Hence, defining $\mathbf{B}$ to be the matrix associated to the operator $T_n A T_n^* : \mathbb{R}^n \to \mathbb{R}^n$, it holds $\Pi_n A \Pi_n = S_n^* \mathbf{B} S_n$. Moreover, $A \succeq 0$ implies $\mathbf{B} = T_n A T_n^* \succeq 0$.

**2. Proving** $\left\{\sum_{1\leq i,j\leq n}\mathbf{B}_{i,j}\phi(x_i)\phi(x_j)^\top, : \mathbf{B}\in\mathbb{R}^{n\times n}, \mathbf{B}\succeq 0\right\} \subset \mathcal{S}_n(\mathcal{H})_+$. Let $\mathbf{B}\in\mathbb{R}^{n\times n}$, $\mathbf{B}\succeq 0$. Since $\mathbf{B}\succeq 0$, $A := S_n^*\mathbf{B}S_n \succeq 0$. Since $S_n^*$ has its range included in $\mathcal{H}_n$, $\Pi_n S_n^* = S_n^*$. Thus, $\Pi_n A\Pi_n = A$ and hence $A\in\mathcal{S}_n(\mathcal{H})_+$.

The second statement comes from the definition of $f_A(x)$. Indeed assume $A\in\mathcal{S}_n(\mathcal{H})_+$. By definition, $f_A(x)=\phi(x)^\top A\phi(x)$. Moreover, by the previous point, there exists $\mathbf{B}\in\mathbb{R}^{n\times n}$, $\mathbf{B}\succeq 0$ such that $A=\sum_{1\leq i,j\leq n}\mathbf{B}_{i,j}\phi(x_i)\phi(x_j)^\top$. Combining these two facts yields:

$$\forall x\in\mathcal{X},\ f_A(x)=\sum_{1\leq i,j\leq n}\mathbf{B}_{i,j}\phi(x)^\top\phi(x_i)\ \phi(x_j)^\top\phi(x)=\sum_{1\leq i,j\leq n}\mathbf{B}_{i,j}k(x,x_i)\ k(x,x_j).$$

The last equality comes from the definition $k(x,\tilde{x})=\phi(x)^\top\phi(\tilde{x})$. $\qquad\square$

*Proof of Thm. 7.* Under the assumptions of Thm. 7, one satisfies the assumptions of Proposition 7. Thus, Eq. (5) has a solution $A_*$ which is in $\mathcal{S}_n(\mathcal{H})_+$. Now applying Lemma 3, $A_*$ can be written in the form $A_*=\sum_{i,j}\mathbf{B}_{i,j}\phi(x_i)\phi(x_j)^\top$ for $\mathbf{B}\in\mathbb{R}^{n\times n}$, $\mathbf{B}\succeq 0$, and hence

$$\forall x\in\mathcal{X},\ f_{A_*}(x)=\sum_{i,j}\mathbf{B}_{i,j}k(x,x_i)k(x,x_j).$$

Uniqueness in the case where $\Omega$ is of the form Eq. (6) with $\lambda_2>0$ comes from the fact that the loss function is strongly convex in this case, and thus the minimizer is unique. $\qquad\square$

## B.4 Proof of Proposition 3

Recall the definitions of $S_n:\mathcal{H}\to\mathbb{R}^n$ and its adjoint $S_n^*:\mathbb{R}^n\to\mathcal{H}$ :

$$\forall h,\ S_n(h)=\left(h^\top\phi(x_i)\right)_{1\leq i\leq n},\ \forall\alpha\in\mathbb{R}^n,\ S_n^*\alpha=\sum_{i=1}^n\alpha_i\phi(x_i).$$

Note that the kernel matrix $\mathbf{K}=(k(x_i,x_j))_{1\leq i,j\leq n}$ can also be written as $\mathbf{K}=S_nS_n^*$.
Let $r$ be the rank of $\mathbf{K}$ and $\mathbf{V}\in\mathbb{R}^{r\times n}$ be a matrix such that

$$\mathbf{V}^\top\mathbf{V}=\mathbf{K}.$$

Note that $\mathbf{V}$ is of rank $r$ and hence $\mathbf{V}\mathbf{V}^\top$ is invertible, making the following definition of $O_n:\mathbb{R}^r\to\mathcal{H}$ valid:

$$O_n=S_n^*\mathbf{V}^\top(\mathbf{V}\mathbf{V}^\top)^{-1}.$$

The following result holds :

**Lemma 4.** $O_nO_n^*=\Pi_n$ *and* $O_n^*O_n=I_r$.

*Proof.* Using the fact that $\mathbf{V}^\top\mathbf{V}=\mathbf{K}=S_nS_n^*$, we have

$$O_n^*O_n=(\mathbf{V}\mathbf{V}^\top)^{-1}\mathbf{V}S_nS_n^*\mathbf{V}^\top(\mathbf{V}\mathbf{V}^\top)^{-1}=(\mathbf{V}\mathbf{V}^\top)^{-1}\mathbf{V}\mathbf{V}^\top\mathbf{V}\mathbf{V}^\top(\mathbf{V}\mathbf{V}^\top)^{-1}=I_r.$$

Now let us show that $O_nO_n^*=\Pi_n$. First of all, $\tilde{\Pi}_n := O_nO_n^*$ is self adjoint and is a projection operator since $\tilde{\Pi}_n^2 = O_n(O_n^*O_n)O_n^* = O_nO_n^* = \tilde{\Pi}_n$ by the previous point. Moreover, its range is included in $\text{span}(\phi(x_i))_{1\leq i\leq n}$ since $O_n=S_n^*\tilde{O}_n$ for a certain $\tilde{O}_n$ and the range of $S_n^*$ is $\text{span}(\phi(x_i))_{1\leq i\leq n}$. Finally since the rank of $S_n^*$ is also the rank of $S_nS_n^*$ which is $r$, we deduce that the range of $\text{span}(\phi(x_i))_{1\leq i\leq n}$ is of dimension $r$ and hence, since $O_n^*O_n=I_r$ implies that $O_nO_n^*$ is of rank $r$, putting things together, $\tilde{\Pi}_n=\Pi_n$. $\qquad\square$

**Remark 6** (Constructing $\mathbf{V}$). *In the case where the kernel matrix $\mathbf{K}$ is full rank, $\mathbf{V}\in\mathbb{R}^{n\times n}$ and is invertible, and $O_n$ can be simply written $S_n^*\mathbf{V}^{-1}$.*
*In the case where the kernel matrix $\mathbf{K}$ is not full-rank, we build $\mathbf{V}$ as $\mathbf{V}=\mathbf{\Sigma}^{1/2}\mathbf{U}^\top$, where $\mathbf{\Sigma}\in\mathbb{R}^{r\times r}$ is diagonal and $\mathbf{U}\in\mathbb{R}^{n\times r}$ is unitary and correspond to the* economy eigendecomposition *of $\mathbf{K}$ where $r$ is the rank of $\mathbf{K}$, i.e., $\mathbf{K}=\mathbf{U}\mathbf{\Sigma}\mathbf{U}^\top$.*

Consider the following generalization of the finite dimensional model proposed in Eq. (8) in the case where $\mathbf{K}$ is not necessarily full rank :

$$\tilde{f}_{\mathbf{A}}(x) = \Phi(x)^\top \mathbf{A} \Phi(x), \qquad \mathbf{A} \in \mathbb{R}^{r \times r}, \ \mathbf{A} \succeq 0, \tag{8}$$

where $\Phi : \mathcal{X} \mapsto \mathbb{R}^r$ is defined as $\Phi(x) = O_n^* \phi(x) = (\mathbf{V}\mathbf{V}^\top)^{-1} \mathbf{V} v(x)$, where $v(x) = (k(x_i, x))_{1 \le i \le n} \in \mathbb{R}^n$.

We are now ready to prove Proposition 3.

*Proof of Proposition 3.* Recall

$$\min_{\mathbf{A} \succeq 0} L(\tilde{f}_{\mathbf{A}}(x_1), \ldots, \tilde{f}_{\mathbf{A}}(x_n)) + \Omega(\mathbf{A}). \tag{9}$$

The fact that Eq. (9) has a solution, and that this solution is unique if $\lambda_2 > 0$ and $L$ is convex can be seen as a simple consequence of Thm. 7 in the case where the model considered is the finite dimensional model defined in Eq. (8). Let us now prove the other part of the proposition.

Start by noting that with our definition of $O_n$, for all $\mathbf{A} \in \mathbb{R}^{r \times r}$, $\mathbf{A} \succeq 0$,

$$f_{O_n \mathbf{A} O_n^*} = \tilde{f}_{\mathbf{A}}. \tag{a}$$

Moreover,

$$\left\{ O_n \mathbf{A} O_n^* \ : \ \mathbf{A} \in \mathbb{R}^{r \times r}, \ \mathbf{A} \succeq 0 \right\} = \mathcal{S}_n(\mathcal{H})_+. \tag{b}$$

Finally, since $O_n$ is an isometry which implies $\Omega(O_n \mathbf{A} O_n^*) = \Omega(\mathbf{A})$ and by Eq. (a), for any $\mathbf{A} \in \mathcal{S}(\mathbb{R}^n)_+$, it holds :

$$L(f_{O_n \mathbf{A} O_n^*}(x_1), ..., f_{O_n \mathbf{A} O_n^*}(x_n)) + \Omega(O_n \mathbf{A} O_n^*) = L(\tilde{f}_{\mathbf{A}}(x_1), ..., \tilde{f}_{\mathbf{A}}(x_n)) + \Omega(\mathbf{A}). \tag{c}$$

Now combining Eq. (c) and Eq. (b), any solution $\mathbf{A}_*$ to Eq. (9) corresponds to a solution $A_* \in \operatorname{argmin}_{A \in \mathcal{S}_n(\mathcal{H})_+} L(f_A(x_1), ..., f_A(x_n)) + \Omega(A)$, where $A_* = O_n \mathbf{A}_* O_n^*$. Now using Eq. (16) in the proof of Proposition 7, we see that $A_*$ is also a minimizer of Eq. (5) hence the result.

Note that the fact that the condition number of the problem, if it exists, is preserved because $O_n$ is an isometry. $\qquad \square$

## B.5  Proof of Thm. 2 and algorithmic consequence.

In this section, we prove Thm. 2 and explain how to derive an efficient algorithm to solve it in certain cases.

Let us start by proving the following lemma.

**Lemma 5.** *Let $\lambda_1, \lambda_2 \ge 0$ and assume $\lambda_2 > 0$. Let $\Omega_+$ be defined on $\mathcal{S}(\mathbb{R}^r)$ as follows :*

$$\Omega_+(A) = \begin{cases} \lambda_1 \|A\|_\star + \frac{\lambda_2}{2} \|A\|_F^2 & \text{if } A \succeq 0; \\ +\infty & \text{otherwise} . \end{cases}$$

*Then $\Omega_+$ is a closed convex function, and its Fenchel conjugate is given for any $B \in \mathcal{S}(\mathbb{R}^r)$ by the formula:*

$$\Omega_+^*(B) = \frac{1}{2\lambda_2} \left\| [B - \lambda_1 I]_+ \right\|_F^2 .$$

*Moreover, $\Omega_+$ is differentiable at every point, and is $1/\lambda_2$ smooth. Its gradient is given by:*

$$\nabla \Omega_+^*(B) = \frac{1}{\lambda_2} [B - \lambda_1 I]_+ .$$

*Proof.* Write

$$\Omega_+(A) = \iota_{\mathcal{S}(\mathbb{R}^r)_+} + \lambda_1 \|A\|_\star + \frac{\lambda_2}{2} \|A\|_F^2 .$$

Here, $\iota_C$ stands for the characteristic function of the convex set $C$, i.e. $\iota_C(x) = 0$ if $x \in C$ and $+\infty$ otherwise. Since $\|\cdot\|_F^2$ and $\|\cdot\|_\star$ are both convex, continuous, and real valued, and since $\iota_{\mathcal{S}(\mathbb{R}^r)_+}$ is closed since $\mathcal{S}(\mathbb{R}^r)_+$ is a closed non-empty convex subset of $\mathcal{S}(\mathbb{R}^r)$, this shows that $\Omega_+$ is indeed convex and closed. Note that it is continuous on its domain $\mathcal{S}(\mathbb{R}^r)_+$. Moreover, it is strongly convex since $\lambda_2 > 0$. Fix $B \in \mathcal{S}(\mathbb{R}^r)$ and consider the problem

$$\sup_{A \in \mathcal{S}(\mathbb{R}^r)} \mathrm{Tr}(AB) - \Omega_+(A) = \sup_{A \succeq 0} \mathrm{Tr}(A(B - \lambda_1 I)) - \frac{\lambda_2}{2}\|A\|_F^2$$

Since $\Omega_+$ is strongly convex, we know there exists a unique solution to this problem.

Note that $A_* = \mathrm{argmax}\, \mathrm{Tr}(AB) - \Omega_+(A)$ if and only if

$$A_* = \mathrm{argmin}_{A \in \mathcal{S}(\mathbb{R}^r)_+} \frac{1}{2} \left\| \left( A - \frac{1}{\lambda_2}(B - \lambda_1 I) \right) \right\|^2 .$$

That is $A_*$ is the orthogonal projection of $\frac{B - \lambda_1 I}{\lambda_2}$ on $\mathcal{S}(\mathbb{R}^r)_+$ for the Frobenius scalar product. Hence, $A_* = \left[ \frac{B - \lambda_1 I}{\lambda_2} \right]_+$.

Here, for any symetric matrix $C$, we denote with $[C]_+$ resp $[C]_-$ its positive resp negative part. Given an eigendecomposition $C = U\Sigma U^T$ with $\Sigma$ diagonal, they are defined by $[C]_+ = U\max(0, \Sigma)U^T$ and $[C]_- = U\max(0, -\Sigma)U^T$. Hence, the Fenchel conjugate of $\Omega_+$ is given by

$$\Omega_+^*(B) = \frac{1}{2\lambda_2} \left\| [B - \lambda_1 I]_+ \right\|_F^2 .$$

Consider $\omega_+^* : \sigma \in \mathbb{R} \mapsto \max(0, \sigma^2) \in \mathbb{R}$. $\omega_+^*$ is 1-smooth and differentiable, and $(\omega_+^*)'(\sigma) = \max(0, \sigma)$. Hence, the function

$$B \mapsto \mathrm{Tr}(\omega_+^*(B)) = \|[B]_+\|_F^2$$

is differentiable and 1-smooth, with differential given by the spectral function $(\omega_+^*)'(B) = [B]_+$. Hence, $\Omega_+$ is differentiable and $\nabla\Omega_+^*(B) = \frac{1}{\lambda_2}[B - \lambda_1 I]_+$, and is $1/\lambda_2$ smooth. $\square$

**Theorem 8** (Convex dual problem). *Let $L : \mathbb{R}^n \to \mathbb{R} \cup \{+\infty\}$ be convex closed function and $L^*$ be the Fenchel conjugate of $L$ (see [BV04] for the definition of closed and of the dual conjugate). Assume $\Omega$ is of the form Eq. (6). Assume there exists $\mathbf{A} \in \mathbb{R}^{r \times r}$, $\mathbf{A} \succeq \mathbf{0}$ such that $L$ is continuous in $(\tilde{f}_\mathbf{A}(x_i))_{1 \leq i \leq n}$.*

*Then the problem in Eq. (9) has the following dual formulation,*

$$\sup_{\alpha \in \mathbb{R}^n} -L^*(\alpha) - \frac{1}{2\lambda_2}\|[\mathbf{V}\,\mathrm{Diag}(\alpha)\mathbf{V}^\top + \lambda_1\mathbf{I}]_-\|_F^2, \tag{10}$$

*and this supremum is atteined. Let $\alpha^* \in \mathbb{R}^n$ be a solution of (10). Then, the solution of (5) is obtained via (7), with $\mathbf{B} \in \mathbb{R}^{r \times r}, \mathbf{B} \succeq 0$ as*

$$\mathbf{B} = \mathbf{V}^\top(\mathbf{V}\mathbf{V}^\top)^{-1} \left( \frac{1}{\lambda_2} \left[ \mathbf{V}\,\mathrm{Diag}(\alpha_*)\mathbf{V}^\top + \lambda_1 I \right]_- \right) (\mathbf{V}\mathbf{V}^\top)^{-1}\mathbf{V}. \tag{11}$$

*Proof of Thm. 8.* We apply theorem 3.3.1 of [BL10] with the following parameters (on le left, the ones in theorem 3.3.1 of [BL10] and on the right the ones by which we replace them).

| | |
|---|---|
| $\mathbf{E}$ | $\mathcal{S}(\mathbb{R}^r)$ |
| $\mathbf{Y}$ | $\mathbb{R}^n$ |
| $A : \mathbf{E} \to \mathbf{Y}$ | $R : \mathbf{A} \in \mathcal{S}(\mathbb{R}^r) \mapsto (\tilde{f}_\mathbf{A}(x_1), ..., \tilde{f}_\mathbf{A}(x_n)) \in \mathbb{R}^n$ |
| $f : \mathbf{E} \to ]-\infty, +\infty]$ | $\Omega_+ : \mathcal{S}(\mathbb{R}^r) \to ]-\infty, +\infty]$ |
| $g : \mathbf{Y} \to ]-\infty, +\infty]$ | $L : \mathbb{R}^n \to ]-\infty, +\infty]$ |
| $p = \inf_{x \in \mathbf{E}} g(Ax) + f(x)$ | $p = \inf_{\mathbf{A} \in \mathcal{S}(\mathbb{R}^r)} L(\tilde{f}_\mathbf{A}(x_1), ..., \tilde{f}_\mathbf{A}(x_n)) + \Omega_+(\mathbf{A})$ |
| $d = \sup_{\phi \in \mathbf{Y}} -g^*(\phi) - f^*(-A^*\phi)$ | $d = \sup_{\alpha \in \mathbb{R}^n} -L^*(\alpha) - \Omega_+^*(-R^*(\alpha))$ |

Indeed, for all $1 \leq i \leq n$, if $\Phi$ is defined in Eq. (8), $\Phi(x_i) = \mathbf{V}e_i$ and thus $\tilde{f}_{\mathbf{A}}(x_i) = \Phi(x_i)^\top \mathbf{A} \Phi(x_i) = e_i^\top (\mathbf{V}^\top \mathbf{A} \mathbf{V}) e_i$. Thus, for any $\mathbf{A} \in \mathcal{S}(\mathbb{R}^r)$, $R(\mathbf{A}) := \left(\tilde{f}_{\mathbf{A}}(x_i)\right)_{1 \leq i \leq n} = \text{Diag}(\mathbf{V}^\top \mathbf{A} \mathbf{V})$. The following properties are satisfied :

- $L$ is lower semi-continuous, convex and bounded below hence closed (see [BL10]);
- similarly, $\Omega_+$ is a non negative closed convex function, with dual $\Omega_+^*$ given in Lemma 5 which is differentiable and smooth;
- $\text{dom}(\Omega_+) = \mathcal{S}(\mathbb{R}^n)_+$ ;
- $R$ is linear, and for any $\alpha \in \mathbb{R}^n$, it holds $R^*\alpha = \mathbf{V}\text{Diag}(\alpha)\mathbf{V}^\top$;
- The dual $d$ can therefore be re-expressed as Eq. (10), using the expressions for $\Omega_+^*$ and $R^*$ :

$$\sup_{\alpha \in \mathbb{R}^n} -L^*(\alpha) - \frac{1}{2\lambda_2} \left\| \left[\mathbf{V}\text{Diag}(\alpha)\mathbf{V}^\top + \lambda_1 I\right]_-\right\|_F^2 \tag{10}$$

- Assume there exists $\mathbf{A} \in \mathbb{R}^{r \times r}$, $\mathbf{A} \succeq \mathbf{0}$ such that $L$ is continuous in $(\tilde{f}_{\mathbf{A}}(x_i))_{1 \leq i \leq n}$. Then there exists a point of continuity of $g$ such which is also in $R \, \text{dom} \, f$, hence the assumption of theorem 3.3.1 of [BL10] is satisfied.

Applying theorem 3.3.1 of [BL10], the following properties hold:

- $d = p$,
- $d$ is atteined for a certain $\alpha_* \in \mathbb{R}^n$. Indeed, there exists $\mathbf{A} \in \text{dom} \, \Omega_+$ such that $R(\mathbf{A}) \in \text{dom}(L)$. Thus , $L(R(\mathbf{A})) + \Omega_+(\mathbf{A}) < +\infty$ and hence $d < +\infty$. Moreover, since $L$ and $\Omega_+$ are lower bounded, this shows that $d$ is lower bounded and hence $d > -\infty$. Hence $d$ is finite and thus is atteined by theorem 3.3.1.

Now using Exercise 4.2.17 of [BL10] since $L$ and $\Omega_+$ are closed convex and since $\Omega_+^*$ is differentiable, we see that the optimal solution of the primal problem $\mathbf{A}_*$ is given by the following formula:

$$\mathbf{A}_* = \nabla \Omega_+^*(-R^*\alpha^*) = \frac{1}{\lambda_2}\left[\mathbf{V}\text{Diag}(\alpha_*)\mathbf{V}^\top + \lambda_1 I\right]_- .$$

Thus, for any $x \in \mathcal{X}$, using the definition of $\Phi(x)$, it holds

$$\tilde{f}_{\mathbf{A}}(x) = \Phi(x)^\top \mathbf{A}_* \Phi(x) = v(x)^\top \mathbf{V}^\top (\mathbf{V}\mathbf{V}^\top)^{-1}\left(\frac{1}{\lambda_2}\left[\mathbf{V}\text{Diag}(\alpha_*)\mathbf{V}^\top + \lambda_1 I\right]_-\right)(\mathbf{V}\mathbf{V}^\top)^{-1}\mathbf{V}v(x).$$

Thus, setting

$$\mathbf{B} = \mathbf{V}^\top (\mathbf{V}\mathbf{V}^\top)^{-1}\left(\frac{1}{\lambda_2}\left[\mathbf{V}\text{Diag}(\alpha_*)\mathbf{V}^\top + \lambda_1 I\right]_-\right)(\mathbf{V}\mathbf{V}^\top)^{-1}\mathbf{V},$$

it holds $\tilde{f}_{\mathbf{A}}(x) = v(x)^\top \mathbf{B}v(x)$. Since $v(x) = (k(x, x_i))_{1 \leq i \leq n} \in \mathbb{R}^n$, this shows the result. In particular, note that when $\mathbf{V}$ is invertible (i.e. when $\mathbf{K}$ is full rank) then the equation above is exactly Eq. (11), since $\mathbf{V}^\top(\mathbf{V}\mathbf{V}^\top)^{-1} = \mathbf{V}^{-1}$.

$\square$

*Proof of Thm. 2.* It is a direct consequence of the previous theorem.

$\square$

Note that the conditions of theorem Thm. 2 are satisfied in many interesting cases, such as the ones described in the following proposition.

**Proposition 8.** *Assume one of the following conditions is satisfied :*

- *(i)* $\mathrm{dom}(L) = \mathbb{R}^n$;

- *(ii)* $\mathbb{R}^n_{++} \subset \mathrm{dom}(L)$ *and* $k(x_i, x_i) > 0$ *for all* $1 \leq i \leq n$

- *(iii)* $\mathbf{K}$ *is full rank and there exists a continuity point* $\alpha_0$ *of* $L$ *such that* $\alpha_0 \in \mathbb{R}^n_+$.

*Then there exists* $\mathbf{A} \in \mathcal{S}(\mathbb{R}^n)_+$ *such that* $L$ *is continuous in* $(\tilde{f}_{\mathbf{A}}(x_1), ..., \tilde{f}_{\mathbf{A}}(x_n))$.

*Proof.* Let us prove these points.

- if $\mathrm{dom}(L) = \mathbb{R}^n$, since $L$ is convex, $L$ is continuous everywhere. Taking $\mathbf{A} = \mathbf{0}$, the result holds.

- if $k(x_i, x_i) > 0$ for all $i > 0$, then taking $\mathbf{A} = I_r$, we have $(\tilde{f}_{\mathbf{A}}(x_i))_{1 \leq i \leq n} = (k(x_i, x_i))_{1 \leq i \leq n} \in \mathbb{R}^n_{++}$. Since $\mathbb{R}^n_{++} \subset \mathrm{dom}(L)$ and $\mathbb{R}^n_{++}$ is open, $L$ is continuous on $\mathbb{R}^n_{++}$ and hence, $\mathbf{A}$ satisfies the desired property.

- Let $\alpha_0$ be a continuity point of $L$ in $\mathbb{R}^n_+$. If we assume $\mathbf{K}$ is full rank, then in particular, $\mathbf{V} \in \mathbb{R}^{n \times n}$ is of rank $n$ and invertible. Thus, there exists $\mathbf{A} \in \mathcal{S}(\mathbb{R}^r)_+$ such that

$$\mathbf{V}^\top \mathbf{A} \mathbf{V} = \mathrm{Diag}(\alpha_0) \implies (\tilde{f}_{\mathbf{A}}(x_i))_{1 \leq i \leq n} = \alpha_0.$$

$\square$

**Discussion on how to solve Eq. (10)** Proximal splitting methods can be applied to solve Eq. (10) such as FISTA [BT09], provided the proximal operator of $L^*$ can be computed (see [PB14] for the definition of the proximal operator). Indeed, Eq. (10) can be written as

$$\min_{\alpha \in \mathbb{R}^n} F(\alpha) = f(\alpha) + g(\alpha), \qquad f(\alpha) = \Omega^*_+(-\mathbf{V}\,\mathrm{Diag}(\alpha)\mathbf{V}^\top), \; g(\alpha) = L^*(\alpha).$$

where $\Omega^*_+$ has been defined in Lemma 5 and has been shown to be smooth and differentiable. Thus, since $\alpha \mapsto \mathbf{V}\,\mathrm{Diag}(\alpha)\mathbf{V}^\top$ is linear, $f$ is smooth and differentiable. Moreover, one can have access to the gradient of $f$ by performing an eigenvalue decomposition of $\mathbf{V}\,\mathrm{Diag}(\alpha)\mathbf{V}^\top$ whose complexity is bounded above by $\mathcal{O}(r^3)$. Thus, one can apply one of the algorithms in section 4 of [BT09] in order to compute an optimal solution to Eq. (10). Moreover, a bound on the performance of the algorithm is given in theorem 4.4 of this same work. Note that if $L$ is of the form $L(\alpha) = \sum_{i=1}^n \ell_i(\alpha_i)$, it suffices to be able to compute the proximal operator of the $\ell_i$ to get a proximal operator for $L^*$ (see [PB14]).

## B.6 Proof and additional discussion of Thm. 3

We recall the notion of universality [MXZ06], in particular *cc-universality* [SFL11], here explicited in the context of non-negative functions. A set $\mathcal{F}$ is a *universal approximator* for non-negative functions on $\mathcal{X}$ if, for any compact subset $\mathcal{Z}$ of $\mathcal{X}$, we have that the set $\mathcal{F}|_{\mathcal{Z}}$ of restrictions on $\mathcal{Z}$, defined as $\mathcal{F}|_{\mathcal{Z}} = \{f|_{\mathcal{Z}} \mid f \in \mathcal{F}\}$, is dense in the set $C^+(\mathcal{Z})$ of non-negative continuous functions over $\mathcal{Z}$ in the maximum norm. In the following theorem we prove the cc-universality of the proposed model

**Theorem 9.** *Let* $\mathcal{X}$ *be a locally compact Hausdorff space,* $\mathcal{H}$ *a separable Hilbert space and* $\phi : \mathcal{X} \to \mathcal{H}$ *a cc-universal feature map. Let* $\| \cdot \|_\circ$ *be a norm for* $\mathcal{S}(\mathcal{H})$ *such that* $\| \cdot \|_\star \unrhd \| \cdot \|_\circ$. *Then* $\mathcal{F}^\circ_\phi$ *is a cc-universal approximator for the non-negative functions on* $\mathcal{X}$.

*Proof.* Proving that the proposed model is a cc-universal approximator for non-negative functions, is equivalent to require that given a compact set $\mathcal{Z} \subseteq \mathcal{X}$, a non-negative function $g : \mathcal{Z} \to \mathbb{R}_+$ and $\epsilon > 0$, there exists $f_{A_{g,\mathcal{Z},\epsilon}} \in \mathcal{F}^\circ_\phi$ such that $\|g - f_{A_{g,\mathcal{Z},\epsilon}}\|_{C(Z)} \leq \epsilon$. In particular, let $Q = 2\|g\|^{1/2}_{C(\mathcal{Z})} + \epsilon^{1/2}$, since $\phi$ is *cc*-universal, given $\mathcal{Z}, g, \epsilon$, there exists $w_{\sqrt{g},\mathcal{Z},\frac{\epsilon}{Q}}$ such that $\|\sqrt{g} - \phi(\cdot)^\top w_{\sqrt{g},\mathcal{Z},\frac{\epsilon}{Q}}\|_{C(Z)} \leq \frac{\epsilon}{Q}$. Define $A_{g,\mathcal{Z},\epsilon} = w_{\sqrt{g},\mathcal{Z},\frac{\epsilon}{Q}} \otimes w_{\sqrt{g},\mathcal{Z},\frac{\epsilon}{Q}}$. Note that for any $x \in \mathcal{X}$,

$$f_{A_{g,\mathcal{Z},\epsilon}}(x) = \phi(x)^\top A_{g,\mathcal{Z},\epsilon}\phi(x) = \phi(x)^\top \left( w_{\sqrt{g},\mathcal{Z},\frac{\epsilon}{Q}} \otimes w_{\sqrt{g},\mathcal{Z},\frac{\epsilon}{Q}} \right) \phi(x) = (\phi(x)^\top w_{\sqrt{g},\mathcal{Z},\frac{\epsilon}{Q}})^2.$$

$$(17)$$

Then, by denoting with $h(x) = \sqrt{g(x)} - \phi(x)^\top w_{\sqrt{g}, \mathcal{Z}, \frac{\epsilon}{Q}}$, we have

$$\|g - f_{A_g, \mathcal{Z}, \epsilon}\|_{C(Z)} = \sup_{x \in \mathcal{Z}} |g(x) - (\phi(x)^\top w_{\sqrt{g}, \mathcal{Z}, \frac{\epsilon}{Q}})^2| \tag{18}$$

$$= \sup_{x \in \mathcal{Z}} \left| \left( \sqrt{g(x)} - \phi(x)^\top w_{\sqrt{g}, \mathcal{Z}, \frac{\epsilon}{Q}} \right) \left( \sqrt{g(x)} + \phi(x)^\top w_{\sqrt{g}, \mathcal{Z}, \frac{\epsilon}{Q}} \right) \right| \tag{19}$$

$$= \sup_{x \in \mathcal{Z}} |h(x)(2\sqrt{g(x)} - h(x))| \tag{20}$$

$$\leq \|h\|_{C(\mathcal{Z})} (2\|\sqrt{g}\|_{C(\mathcal{Z})} + \|h\|_{C(\mathcal{Z})}) \tag{21}$$

$$\leq \frac{\epsilon}{Q} \left( 2\|g\|_{C(\mathcal{Z})}^{1/2} + \frac{\epsilon}{Q} \right) \leq \epsilon. \tag{22}$$

The last step is due to the fact that $\epsilon/Q \leq \sqrt{\epsilon}$, then $2\|g\|_{C(\mathcal{Z})}^{1/2} + \frac{\epsilon}{Q} \leq Q$. $\qquad\square$

## B.7 Proof and additional discussion of Thm. 4

In Thm. 10, stated below, we prove that $\mathcal{E}_\phi \subseteq \mathcal{F}_\phi$ under the very general assumption that $\mathcal{G}_\phi$ is a multiplication algebra, i.e.. if $\mathcal{G}_\phi$ is closed under pointwise product of the functions. In Thm. 11 we specify this result when $\mathcal{G}_\phi$ is a Sobolev space, proving that $\mathcal{E}_\phi \subsetneq \mathcal{F}_\phi^\circ$. Thm. 4 is a direct consequence of the latter theorem.

**General result when $\mathcal{G}_\phi$ is a multiplication algebra.** First we endow $\mathcal{G}_\phi$ with a Hilbertian norm. Define $\| \cdot \|_{\mathcal{G}_\phi}$ as $\|f_w\|_{\mathcal{G}_\phi} = \|w\|_\mathcal{H}$, for any $w \in \mathcal{H}$.

**Definition 3.** *$\mathcal{G}_\phi$ is a multiplication algebra, when there exists a constant $C$ such that the unit function $u : \mathcal{X} \to \mathbb{R}$ that maps $x \mapsto 1$ for any $x \in \mathcal{X}$ is in $\mathcal{G}_\phi$ and*

$$\|f \cdot g\|_{\mathcal{G}_\phi} \leq C\|f\|_{\mathcal{G}_\phi}\|g\|_{\mathcal{G}_\phi}, \qquad \forall\, f, g \in \mathcal{G}_\phi, \tag{23}$$

*where we denote by $f \cdot g$ the pointwise multiplication, i.e., $(f \cdot g)(x) = f(x)g(x)$ for all $x \in \mathcal{X}$.*

**Remark 7** (Renormalizing the constant). *Note that when $\mathcal{G}_\phi$ is a multiplication algebra for a constant $C$, it is always possible to define an equivalent norm $\| \cdot \|'_{\mathcal{G}_\phi}$ as $\| \cdot \|'_{\mathcal{G}_\phi} = C\| \cdot \|_{\mathcal{G}_\phi}$ for which $\mathcal{G}_\phi$ is a multiplication algebra with constant $1$.*

**Theorem 10** (General version when $\mathcal{G}_\phi$ is an algebra). *Let $\| \cdot \|_\star \unrhd \| \cdot \|_\circ$. Let $\mathcal{X}$ be a compact space and $\phi$ be a bounded continuous map such that $\mathcal{G}_\phi$ is a multiplication algebra, then $\mathcal{E}_\phi \subseteq \mathcal{F}_\phi^\circ$.*

*Proof.* Let $g \in \mathcal{E}_\phi$ and take $f \in \mathcal{G}_\phi$ such that $g(x) = e^{f(x)}$ for all $x \in \mathcal{X}$. First we prove that $\mathcal{E}_\phi \subseteq \mathcal{F}_\phi^\circ$. With this goal, first we prove that $\sqrt{g} \in \mathcal{G}_\phi$ and then we construct a rank one positive operator such that $f_{A_g}(x) = g(x)$ for every $x \in \mathcal{X}$. We start noting that, given $f \in \mathcal{G}_\phi$ and $t \in \mathbb{N}$, $f^t$ defined by $f \cdot f^{t-1}$ for $t \in \mathbb{N}$ satisfies $f^t \in \mathcal{G}_\phi$, with $\|f^t\|_{\mathcal{G}_\phi} \leq C^t\|f\|_{\mathcal{G}_\phi}^t$, by repeated application of the Eq. (23). Moreover note that the function $s = \sum_{t \in \mathbb{N}} \frac{1}{2^t t!} f^t$, satisfies $s \in \mathcal{G}_\phi$, indeed

$$\|s\|_{\mathcal{G}_\phi} \leq \sum_{t \in \mathbb{N}} \frac{1}{2^t t!}\|f^t\|_{\mathcal{G}_\phi} \leq \sum_{t \in \mathbb{N}} \frac{1}{2^t t!} C^t \|f\|_{\mathcal{G}_\phi}^t \leq e^{C\|f\|_{\mathcal{G}_\phi}/2}.$$

Moreover $s$ satisfies $s(x) = \sqrt{g(x)}$ for all $x \in \mathcal{X}$, indeed for $x \in \mathcal{X}$ we have

$$s(x) = \phi(x)^\top s = \sum_{t \in \mathbb{N}} \frac{1}{2^t t!} \phi(x)^\top f^t = \sum_{t \in \mathbb{N}} \frac{1}{2^t t!} f^t(x) = e^{f(x)/2} = \sqrt{g(x)}.$$

Now let $A_g = s \otimes s$, we have that $\|A_g\|_\circ \leq \|A_g\|_\star$ by assumption, and $\|A_g\|_\star = \|s\|_{\mathcal{G}_\phi}^2 < \infty$, so the function $f_{A_g} \in \mathcal{F}_\phi^\circ$ and for any $x \in \mathcal{X}$

$$f_{A_g}(x) = \phi(x)^\top A_g \phi(x) = \phi(x)^\top (s \otimes s)\phi(x) = (\phi(x)^\top s)^2 = g(x).$$

Since for any $g \in \mathcal{E}_\phi$ there exists $f_{A_g} \in \mathcal{F}_\phi^\circ$ that is equal to $g$ on their domain of definition, we have that $\mathcal{E}_\phi \subseteq \mathcal{F}_\phi^\circ$. $\qquad\square$

Now we are going to specialize the result above for Sobolev spaces.

**Result for Sobolev spaces**   The result below is based on the general result in Thm. 10, however it is possible to do a proof based only on norm inequalities for compositions of functions in Sobolev space (see for example [BM01]). While more technical, this second approach would allow to derive also a more quantitative analysis on the norms of the functions in $\mathcal{G}_\phi$ and $\mathcal{F}_\phi^\circ$. We will leave this for a longer version of this work.

**Theorem 11.** *Let $\|\cdot\|_\star \trianglerighteq \|\cdot\|_\circ$. Let $\mathcal{X} \subseteq \mathbb{R}^d$ and $\mathcal{X}$ compact with locally Lipschitz boundary and let $\mathcal{G}_\phi = W_2^m(\mathcal{X})$. Let $x_0 \in \mathcal{X}$. Then the following holds:*

*(a) $\mathcal{E}_\phi \subsetneq \mathcal{F}_\phi^\circ$.    (b) The function $f_{x_0}(x) = e^{-\|x-x_0\|^{-2}} \in C^\infty(\mathcal{X})$ satisfies $f_{x_0} \in \mathcal{F}_\phi^\circ$ and $f_{x_0} \notin \mathcal{E}_\phi$.*

*Proof.*  First we prove that $\mathcal{E}_\phi \subseteq \mathcal{F}_\phi^\circ$, via Thm. 10, then we show an example of function that is in $\mathcal{F}_\phi^\circ$, but not in $\mathcal{E}_\phi$, obtaining $\mathcal{E}_\phi \subsetneq \mathcal{F}_\phi^\circ$. To apply this result we need first to prove that $\mathcal{G}_\phi = W_2^m(\mathcal{X})$ is a multiplication algebra when $W_2^m(\mathcal{X})$ is a RKHS as in our case.

**Step 1, $m > d/2$.**  First note that $\mathcal{G}_\phi$ satisfies $m > d/2$ since $W_2^m(\mathcal{X})$ admits a representation in terms of a separable Hilbert space $\mathcal{H}$ and a feature map $\phi : \mathcal{X} \to \mathcal{H}$, i.e., it is a *reproducing Kernel Hilbert space* and for the same reason $\|\cdot\|_{\mathcal{G}_\phi}$ is equivalent to $\|\cdot\|_{W_2^m(\mathcal{X})}$ [Wen04].

**Step 2. $\mathcal{G}_\phi$ is a multiplication algebra. Applying Thm. 10.**  Since $\mathcal{G}_\phi = W_2^m(\mathcal{X})$ with $m > d/2$, then it is a multiplication algebra. This result is standard (e.g. see pag. 106 of [AF03] for $m \in \mathbb{N}$ and $\mathcal{X} = \mathbb{R}^d$) and we report it in Lemma 8 in Appendix C. Then we apply Thm. 10 obtaining $\mathcal{E}_\phi \subseteq \mathcal{F}_\phi^\circ$.

**Step 3. Proving that $f_{x_0} \in \mathcal{F}_\phi^\circ$ and not in $\mathcal{E}_\phi$.**  By construction the function $v(x) = e^{-1/(2\|x-x_0\|^2)}$ is in $C^\infty(\mathcal{X})$ and so in $W_2^m(\mathcal{X})$ for any $m \geq 0$. Since $\mathcal{G}_\phi = W_2^m(\mathcal{X})$, then $v \in \mathcal{G}_\phi$, i.e., there exists $w \in \mathcal{H}$ such that $w^\top \phi(\cdot) = v(\cdot)$. Define $A_v = w \otimes w$, then

$$f_{A_v}(x) = \phi(x)^\top A_v \phi(x) = (w^\top \phi(x))^2 = v^2(x) = f_{x_0}(x), \quad \forall x \in \mathcal{X}.$$

Then $f_{x_0} = f_{A_v}$ on $\mathcal{X}$, i.e., $f_{x_0} \in \mathcal{F}_\phi^\circ$. To conclude note that, $f_{x_0}$ does not belong to $\mathcal{E}_\phi$, since $x_0 \in \mathcal{X}$ and $f_{x_0}(x_0) = 0$, while for any $g \in \mathcal{E}_\phi$ we have $\inf_{x \in X} g(x) > 0$. Indeed, we have that for any $f \in \mathcal{G}_\phi$, $\|f\|_{C(\mathcal{X})} = \sup_{x \in \mathcal{X}} |f(x)| < \infty$, since $\mathcal{G}_\phi = W_2^m(\mathcal{X}) \subset C(\mathcal{X})$. Moreover, given $g \in \mathcal{G}_\phi$, and denoting by $f \in \mathcal{G}_\phi$ the function such that $g = e^f$, we have that $\inf_{x \in \mathcal{X}} g(x) \geq e^{-\|f\|_{C(\mathcal{X})}} > 0$. Finally, since $\mathcal{E}_\phi \subseteq \mathcal{F}_\phi^\circ$, but there exists $f_{x_0} \in \mathcal{F}_\phi^\circ$ and not in $\mathcal{E}_\phi$, then $\mathcal{E}_\phi \subsetneq \mathcal{F}_\phi^\circ$. $\qquad\square$

**Proof of Thm. 4.**  This result is a direct application of Thm. 11, since $\mathcal{X} = [-R, R]^d$, with $R \in (0, \infty)$ is a compact set with Lipschitz boundary.

## B.8   Proof of Thm. 5

We recall here the Rademacher complexity and prove Thm. 5. This latter theorem is obtained from the following Thm. 12 that bounds the *empirical Rademacher complexity* introduced below. First we recall that the function class $\mathcal{F}_{\phi,L}^\circ$ is defined as

$$\mathcal{F}_{\phi,L}^\circ = \{f_A \mid A \succeq 0, \|A\|_\circ \leq L\},$$

for a given norm $\|\cdot\|_\circ$ on operators, a feature map $\phi : \mathcal{X} \to \mathcal{H}$ and $L > 0$. Now we define the empirical Rademacher complexity and the Rademacher complexity [BM02]. Given $x_1, \ldots, x_n \in \mathcal{X}$, the empirical Rademacher complexity for a class $\mathcal{F}$ of functions mapping $\mathcal{X}$ to $\mathbb{R}$, is defined as

$$\widehat{R}_n(\mathcal{F}) = 2\mathbb{E} \sup_{f \in \mathcal{F}} \left| \frac{1}{n} \sum_{i=1}^n \sigma_i f(x_i) \right|,$$

where $\sigma_i$ independent Rademacher random variables, i.e., $\sigma_i = -1$ with probability $1/2$ and $+1$ with probability $1/2$ and the expectation is on $\sigma_1, \ldots, \sigma_n$. Let $\rho$ be a probability distribution on $\mathcal{X}$ and $x_1, \ldots, x_n$ sampled independently according to $\rho$. The *Rademacher complexity $R_n(\mathcal{F})$* is defined as

$$R_n(\mathcal{F}) = \mathbb{E}\widehat{R}_n(\mathcal{F}),$$

where the last expectation is on $x_1, \ldots, x_n$. In the following theorem we bound $\widehat{R}_n$.

**Theorem 12.** *Let* $\| \cdot \|_\circ \unrhd \| \cdot \|_F$. *Let* $x_1, \dots, x_n \in \mathcal{X}$, $L \geq 0$.

$$\widehat{R}_n(\mathcal{F}_{\phi,L}^\circ) \leq \frac{2L}{n} \sqrt{\sum_{i=1}^n \|\phi(x_i)\|^4}.$$

*Proof.* Given $f_A \in \mathcal{F}_{\phi,L}^\circ$, since $\| \cdot \|_\circ$ is stronger or equivalent to Hilbert-Schmidt norm, we have that $\|A\|_F \leq \|A\|_\circ \leq L$. Since $A$ is bounded and $\phi(\cdot) \in \mathcal{H}$, by linearity of the trace we have $f_A(x) = \phi(x)^\top A \phi(x) = \mathrm{Tr}(A\, \phi(x) \otimes \phi(x))$ for any $x \in \mathcal{X}$. Then, by linearity of the trace

$$\hat{R}_n(\mathcal{F}_{\phi,L}^\circ) = 2\mathbb{E} \sup_{f \in \mathcal{F}_{\phi,L}^\circ} \left| \frac{1}{n} \sum_{i=1}^n \sigma_i f(x_i) \right| = 2\mathbb{E} \sup_{A \succeq 0, \|A\|_\circ \leq L} \left| \frac{1}{n} \sum_{i=1}^n \sigma_i \phi(x_i)^\top A \phi(x_i) \right| \quad (24)$$

$$= 2\mathbb{E} \sup_{A \succeq 0, \|A\|_\circ \leq L} \left| \frac{1}{n} \sum_{i=1}^n \sigma_i \, \mathrm{Tr}(A\, (\phi(x_i) \otimes \phi(x_i))) \right| \quad (25)$$

$$= 2\mathbb{E} \sup_{A \succeq 0, \|A\|_\circ \leq L} \left| \mathrm{Tr}\left( A\, \left( \frac{1}{n} \sum_{i=1}^n \sigma_i \phi(x_i) \otimes \phi(x_i) \right) \right) \right| \quad (26)$$

Now since $\| \cdot \|_\circ$ is stronger or equivalent to $\| \cdot \|_F$ this means that $\{A \in \mathcal{S}(\mathcal{H}) \mid \|A\|_\circ \leq L\} \subseteq \{A \in \mathcal{S}(\mathcal{H}) \mid \|A\|_F \leq L\}$, then

$$2\mathbb{E} \sup_{A \succeq 0, \|A\|_\circ \leq L} \left| \mathrm{Tr}\left( A\, \left( \frac{1}{n} \sum_{i=1}^n \sigma_i \phi(x_i) \otimes \phi(x_i) \right) \right) \right| \quad (27)$$

$$\leq 2\mathbb{E} \sup_{A \succeq 0, \|A\|_F \leq L} \left| \mathrm{Tr}\left( A\, \left( \frac{1}{n} \sum_{i=1}^n \sigma_i \phi(x_i) \otimes \phi(x_i) \right) \right) \right| \quad (28)$$

$$\leq 2\mathbb{E} \sup_{A \succeq 0, \|A\|_F \leq L} \|A\|_F \left\| \frac{1}{n} \sum_{i=1}^n \sigma_i \phi(x_i) \otimes \phi(x_i) \right\|_F \quad (29)$$

$$\leq 2L\, \mathbb{E} \left\| \frac{1}{n} \sum_{i=1}^n \sigma_i \phi(x_i) \otimes \phi(x_i) \right\|_F. \quad (30)$$

To conclude denote by $\zeta_i$ the random variable $\sigma_i \phi(x_i) \otimes \phi(x_i)$. Then

$$\mathbb{E} \left\| \frac{1}{n} \sum_{i=1}^n \sigma_i \phi(x_i) \otimes \phi(x_i) \right\|_F^2 = \mathbb{E} \left\| \frac{1}{n} \sum_{i=1}^n \zeta_i \right\|_F$$

$$= \mathbb{E} \sqrt{\mathrm{Tr}\left( \left( \frac{1}{n} \sum_{i=1}^n \zeta_i \right)^* \left( \frac{1}{n} \sum_{i=1}^n \zeta_i \right) \right)} = \mathbb{E} \sqrt{\mathrm{Tr}\left( \frac{1}{n^2} \sum_{i,j=1}^n \zeta_i \zeta_j \right)}.$$

By Jensen inequality, the concavity of the square root, and the linearity of the trace

$$\mathbb{E} \sqrt{\mathrm{Tr}\left( \frac{1}{n^2} \sum_{i,j=1}^n \zeta_i \zeta_j \right)} \leq \sqrt{\mathbb{E}\, \mathrm{Tr}\left( \frac{1}{n^2} \sum_{i,j=1}^n \zeta_i \zeta_j \right)} = \sqrt{\frac{1}{n^2} \sum_{i,j=1}^n \mathrm{Tr}(\mathbb{E}\zeta_i \zeta_j)}.$$

Now note that for $i \in \{1, \dots, n\}$, we have $\mathbb{E}_{\sigma_i} \zeta_i = 0$, moreover $\mathbb{E}\sigma_i^2 = \|\phi(x_i)\|^2 \phi(x_i) \otimes \phi(x_i)$. Finally, given $x_1, \dots, x_n$, we have that $\zeta_i$ is independent from $\zeta_j$, when $i \neq j$. Then when $i \neq j$ we have $\mathrm{Tr}(\mathbb{E}\zeta_i \zeta_j) = \mathrm{Tr}((\mathbb{E}_{\sigma_i} \zeta_i)(\mathbb{E}_{\sigma_j} \zeta_j)) = 0$. When $i = j$ we have $\mathrm{Tr}(\mathbb{E}\zeta_i \zeta_j) = \mathrm{Tr}(\mathbb{E}\zeta_i^2) = \|\phi(x_i)\|^4$. So

$$\frac{1}{n^2} \sum_{i,j=1}^n \mathrm{Tr}(\mathbb{E}\zeta_i \zeta_j) = \frac{1}{n^2} \sum_{i=1}^n \|\phi(x_i)\|^4.$$

From which we obtain the desired result. $\qquad\square$

Now we are ready to bound $R_n$ as follows

**Proof Thm. 5.** The proof is obtained by applying Thm. 12 and considering that $\|\phi(x)\|$ is uniformly bounded by $c$ on $\mathcal{X}$. Then

$$R_n(\mathcal{F}) = \mathbb{E}_{x_1,\dots,x_n} \widehat{R}_n(\mathcal{F}) \leq \mathbb{E}_{x_1,\dots,x_n} \frac{2L}{n} \sqrt{\sum_{i=1}^n \|\phi(x_i)\|^4} \leq \frac{2Lc^2}{\sqrt{n}}.$$

$\square$

## B.9 Learning rates

By using a standard argument based on the Rademacher complexity (see [SSBD14] Chapter 26, or [Bac17] paragraph 4.5 and in particular Eq. (13) we can derive the following learning rates for the proposed model. Let the population risk be defined as $R(f) = \mathbb{E}_{x,y}\ell(y, f(x))$ for some $G$-Lipschitz loss function $\ell$ and $\widehat{R}_D$ be the empirical version $\widehat{R}_D(f) = \frac{1}{n}\sum_{i=1}^n \ell(y_i, f(x_i))$ for a given dataset $D$ of $n$ examples. Recall we are given a norm $\|\cdot\|_\circ$ (e.g., Frobenius or nuclear), a feature map $\phi$ and a radius $L$, and that we define the class of estimators $\mathcal{F}_{\phi,L}^\circ := \{f_A \mid \|A\|_\circ \leq L\}$. Denote by $\widehat{f}_{D,L} = \arg\min_{f \in \mathcal{F}_{\phi,L}^\circ} \widehat{R}_D(f)$ the empirical risk minimization solution over the set $\mathcal{F}_{\phi,L}^\circ$, so

$$\mathbb{E}_D R(\widehat{f}_{D,L}) - \inf_{f \in \mathcal{F}_{\phi,L}^\circ} R(f) \leq 2\,\mathbb{E}_D\Big[ \sup_{f \in \mathcal{F}_{\phi,L}^\circ} |R(f) - \widehat{R}_D(f)| \Big] \leq 2G\, \mathcal{R}_n(\mathcal{F}_{\phi,L}^\circ),$$

where $\mathcal{R}_n(\mathcal{F}_{\phi,L}^\circ)$ is the Rademacher complexity of the set $\mathcal{F}_{\phi,L}^\circ$ and is bounded by $\frac{2Lc^2}{\sqrt{n}}$ by Thm. 12. ($c$ is the bounding constant of the kernel, i.e., $c = \sup_{x \in \mathcal{X}} \|\phi(x)\|$).

**Remark 8.** *Note that assuming there exists an operator $A_\star$ with $\|A_\star\|_\circ$ finite (in particular it could be rank-1, i.e. $A_\star = w_\star w_\star^\top$ for some $w_\star$), such that the learning problem is well posed, i.e. $\inf_{f \in C(X)} R(f) = R(f_{A_\star})$, and choosing $L = \|A_\star\|_\circ$, we obtain the learning rate $\mathbb{E}_D R(\widehat{f}_{D,L}) - R(f_{A_\star}) = O(c^2 G \|A_\star\|_\circ/\sqrt{n})$, which is the standard rate for linear models (see [SSBD14] for more details).*

## B.10 Proof of Proposition 4

See Appendix A for the basic technical assumptions on $\mathcal{X}$, $\mathcal{H}$ and $\phi$. In particular $\mathcal{X}$ is Polish and $\phi$ is continuous and uniformly bounded by a constant $c$.

*Proof of Proposition 4.* In the following we will consider integrability and measurability with respect to a measure $dx$ on $\mathcal{X}$. In particular $p : \mathcal{X} \to \mathbb{R}$ is an integrable function on $\mathcal{X}$ with respect to the measure $dx$. Now define $\Psi(x) = p(x)\phi(x)\phi(x)^\top$. We have that $\Psi$ is measurable, since $\phi$ and $p$ are measurable. Since $p$ is integrable, $p$ is finite almost everywhere, and hence $\Psi(x) = p(x)\phi(x)\phi(x)^\top$ is defined and trace class almost everywhere, and satisfies

$$\|\Psi(x)\|_\star = |p(x)|\, \|\phi(x)\|_\mathcal{H}^2 \leq |p(x)|c^2 \text{ almost everywhere.}$$

Since the space of trace class operators is separable, this shows that $\Psi$ is Bochner integrable and thus that the operator $W_p = \int_{x \in \mathcal{X}} \phi(x)\phi(x)^\top p(x)dx$ is well defined and trace class, with trace norm bounded by $\kappa^2 \|p\|_{L^1(\mathcal{X})}$. Moreover, by linearity of the integral, for any $A \in \mathcal{S}(\mathcal{H})$,

$$\mathrm{Tr}(A W_p) = \int_\mathcal{X} \mathrm{Tr}(A\phi(x)\phi(x)^\top)p(x)dx = \int_\mathcal{X} f_A(x)p(x)dx,$$

where the last equality follows from the definition of $f_A$ and the fact that

$$\mathrm{Tr}(A\phi(x)\phi(x)^\top) = \mathrm{Tr}(\phi(x)^\top A\phi(x)) = \phi(x)^\top A\phi(x) = f_A(x).$$

$\square$

**Remark 9** (Extension to more general linear functionals.)**.** *Note that the linearity of the model in $A$ allows to generalize very easily the construction above to any linear functional that we want to apply to the model. This is especially true when the model has a finite dimensional representation as*

Eq. (7), i.e. $f_{\mathbf{B}} = \sum_{ij=1}^{n} \mathbf{B}_{i,j} k(x, x_i) k(x, x_j)$ with $\mathbf{B} \succeq 0$. *In this case, given a linear functional* $\mathcal{L} : C(\mathcal{X}) \to \mathbb{R}$, *we have*

$$\mathcal{L}(f_{\mathbf{B}}) = \sum_{i,j=1}^{n} \mathbf{B}_{i,j} \mathcal{L}(k(x, x_i) k(x, x_j)) = \mathrm{Tr}(\mathbf{B} \mathbf{W}_{\mathcal{L}}),$$

*where* $(\mathbf{W}_{\mathcal{L}})_{i,j} = \mathcal{L}(k(x, x_i) k(x, x_j))$ *for* $i, j = 1, \dots, n$.

## B.11  Proof of Proposition 5

In Appendix B.11 and Appendix B.12, we will use the following notations.

Let $h, p \in \mathbb{N}$ and $\mathcal{H}, \mathcal{H}_1, \mathcal{H}_2$ be separable Hilbert spaces.

- $A = (A_s)_{1 \le s \le p} \in \mathcal{S}(\mathcal{H})^p$ will denote a family of self-adjoint operators;
- Given a feature map $\phi : \mathcal{X} \to \mathcal{H}$ and $A = (A_s)_{1 \le s \le p} \in \mathcal{S}(\mathcal{H})^p$ we will define the function $f_A$ as follows

$$\forall x \in \mathcal{X}, \ f_A(x) = (f_{A_s}(x))_{1 \le s \le p} = \left( \phi(x)^\top A_s \phi(x) \right)_{1 \le s \le p} \in \mathbb{R}^p, \qquad f_A : \mathcal{X} \to \mathbb{R}^p$$

- Given a matrix $C \in \mathbb{R}^{p \times h}$ which corresponds to a list of column vectors $(c^t)_{1 \le t \le h} \in (\mathbb{R}^p)^h$, we define

$$K^C(\mathcal{H}) := \left\{ A = (A_s)_{1 \le s \le p} \in \mathcal{S}(\mathcal{H})^p \ : \ \sum_{s=1}^{p} c_s^t A_s \succeq 0, \ 1 \le t \le h \right\}$$

- For any $A = (A_s)_{1 \le s \le p} \in \mathcal{S}(\mathcal{H}_1)^p$ and any bounded linear operator $L : \mathcal{H}_1 \to \mathcal{H}_2$, $LAL^*$ will be a slight abuse of notation to denote the family $(LA_sL^*)_{1 \le s \le p} \in \mathcal{S}(\mathcal{H}_2)^p$.

*Proof of Proposition 5.* Let $p, h \in \mathbb{N}$ and let $C \in \mathbb{R}^{p \times h}$ be a matrix representing the column vectors $c^1 \dots c^h$.
Let $\mathcal{Y}$ be the polyhedral cone defined by $C$, i.e. $\mathcal{Y} = \left\{ y \in \mathbb{R}^p \ : \ C^\top y \ge 0 \right\}$.
Let $\mathcal{H}$ be a separable Hilbert space and $\phi : \mathcal{X} \to \mathcal{H}$ be a fixed feature map.
With our previous notations, our goal is to prove that for any $A = (A_s)_{1 \le s \le p} \in \mathcal{S}(\mathcal{H})^p$,

$$A \in K^C(\mathcal{H}) \implies \forall x \in \mathcal{X}, \ f_A(x) \in \mathcal{Y}.$$

Assume $A \in K^C(\mathcal{H})$ and let $x \in \mathcal{X}$. By definition, $f_A(x) = (\phi(x)^\top A_s \phi(x))_{1 \le s \le p} \in \mathbb{R}^p$. Hence,

$$C^\top f_A(x) = \left( \sum_{s=1}^{p} c_s^t \phi(x)^\top A_s \phi(x) \right)_{1 \le t \le h} = \left( \phi(x)^\top \left( \sum_{s=1}^{p} c_s^t A_s \right) \phi(x) \right)_{1 \le t \le h}.$$

Since $A \in K^C(\mathcal{H})$, for all $1 \le t \le h$, it holds $\sum_{s=1}^{p} c_s^t A_s \succeq 0$. In particular, this implies $\phi(x)^\top \sum_{s=1}^{p} c_s^t A_s \phi(x) \ge 0$ for all $1 \le t \le h$. Hence

$$C^\top f_A(x) \ge 0 \implies f_A(x) \in \mathcal{Y}.$$

$\square$

## B.12  Proof of Thm. 6

Using the notations of the previous section, the goal of this section is to solve a problem of the form

$$\inf_{A \in K^C(\mathcal{H})} L(f_A(x_1), ..., f_A(x_n)) + \Omega(A), \tag{15}$$

for given $p, h \in \mathbb{N}$, $C \in \mathbb{R}^{p \times h}$, separable Hilbert space $\mathcal{H}$, feature map $\phi : \mathcal{X} \to \mathcal{H}$, regularizer $\Omega$, loss function $L : \mathbb{R}^n \to \mathbb{R} \cup +\infty$ and $x_1, ..., x_n \in \mathcal{X}$.

We start by stating the form of the regularizers we will be using.

**Assumption 2.** *Let $p \in \mathbb{N}$. For any separable Hilbert space $\mathcal{H}$ and any $A = (A_s)_{1 \le s \le p} \in \mathcal{S}(\mathcal{H})^p$, $\Omega$ is of the form*

$$\Omega(A) = \sum_{s=1}^{p} \Omega_s(A_s), \qquad \Omega_s(A_s) = \lambda_{s,1} \|A_s\|_\star + \frac{\lambda_{s,2}}{2} \|A_s\|_F^2,$$

*where $\lambda_{s,1}, \lambda_{s,2} \ge 0$ and $\lambda_{s,1} + \lambda_{s,2} > 0$.*

**Lemma 6** (Properties of $\Omega$). *Let $\Omega$ be a regularizer such that $\Omega$ satisfies Assumption 2. Then $\Omega$ satisfies the following properties.*

(i) *For any separable Hilbert spaces $\mathcal{H}_1, \mathcal{H}_2$ and any linear isometry $O : \mathcal{H}_1 \to \mathcal{H}_2$, i.e., such that $O^*O = I_{\mathcal{H}_1}$, it holds*

$$\forall A \in \mathcal{S}(\mathcal{H}_1)^p, \ \Omega(OAO^*) = \Omega(A).$$

(ii) *For any separable Hilbert space $\mathcal{H}$ and any orthogonal projection $\Pi \in \mathcal{S}(\mathcal{H}_1)$, i.e. satisfying $\Pi = \Pi^*$, $\Pi^2 = \Pi$, it holds*

$$\forall A \in \mathcal{S}(\mathcal{H})^p, \ \Omega(\Pi A \Pi) \le \Omega(A).$$

(iii) *For any finite dimensional Hilbert space $\mathcal{H}_n$, taking $\|A_s\|_{op}$ to be the operator norm on $\mathcal{H}_n$,*

$$\Omega \text{ is continuous,} \qquad \Omega(A) \xrightarrow[\sup_s \|A_s\|_{op} \to +\infty]{} +\infty$$

*Proof.* Note that since

$$\Omega(A) = \sum_{s=1}^{p} \Omega_s(A_s), \qquad \Omega_s(A_s) = \lambda_{s,1} \|A_s\|_\star + \frac{\lambda_{s,2}}{2} \|A_s\|_F^2,$$

where $\lambda_{s,1}, \lambda_{s,2} \ge 0$ and $\lambda_{s,1} + \lambda_{s,2} > 0$, it is actually sufficient to prove the following result.
Let $\lambda_1, \lambda_2 \ge 0$ and assume $\lambda_1 + \lambda_2 > 0$. Let for any $A \in \mathcal{S}(\mathcal{H})$, $\Omega(A) = \lambda_1 \|A\|_\star + \frac{\lambda_2}{2}\|A\|_F^2$. Then the following hold:

(i) For any separable Hilbert spaces $\mathcal{H}_1, \mathcal{H}_2$ and any linear isometry $O : \mathcal{H}_1 \to \mathcal{H}_2$, i.e., such that $O^*O = I_{\mathcal{H}_1}$, it holds

$$\forall A \in \mathcal{S}(\mathcal{H}_1)^p, \ \Omega(OAO^*) = \Omega(A).$$

(ii) For any separable Hilbert space $\mathcal{H}$ and any orthogonal projection $\Pi \in \mathcal{S}(\mathcal{H}_1)$, i.e. satisfying $\Pi = \Pi^*$, $\Pi^2 = \Pi$, it holds

$$\forall A \in \mathcal{S}(\mathcal{H})^p, \ \Omega(\Pi A \Pi) \le \Omega(A).$$

(iii) For any finite dimensional Hilbert space $\mathcal{H}_n$,

$$\Omega \text{ is continuous,} \qquad \Omega(A) \xrightarrow[\|A\|_{op} \to +\infty]{} +\infty,$$

where we denote by $\|\cdot\|_{op}$ the operatorial norm.

**1.** (i) has already been proven in Lemma 2.

**2.** Let us prove (ii). Let $\mathcal{H}$ be a separable Hilbert space, $\Pi$ an orthogonal projection on $\mathcal{H}$ and $A \in \mathcal{S}(\mathcal{H})$.

Using the fact that $\|B\|_\star = \sup_{\|C\|_{op} \le 1} \operatorname{Tr}(BC)$, where $\|C\|_{op}$ denotes the operator norm on $\mathcal{S}(\mathcal{H})$, we have by property of the trace

$$\|\Pi A \Pi\|_\star = \sup_{\|C\|_{op} \le 1} \operatorname{Tr}(\Pi A \Pi C) = \sup_{\|C\|_{op} \le 1} \operatorname{Tr}(A(\Pi C \Pi)).$$

Now since $\|\Pi C \Pi\|_{op} \leq \|C\|_{op} \leq 1$, it holds $\sup_{\|C\|_{op} \leq 1} \operatorname{Tr}(A(\Pi C \Pi)) \leq \sup_{\|C\|_{op} \leq 1} \operatorname{Tr}(AC) = \|A\|_\star$. Thus:

$$\|\Pi A \Pi\|_\star \leq \|A\|_\star.$$

Moreover, since $\Pi \preceq I$, it holds $\Pi A \Pi A \Pi \preceq \Pi A^2 \Pi$. Hence,

$$\|\Pi A \Pi\|_F^2 = \operatorname{Tr}(\Pi A \Pi \Pi A \Pi) \leq \operatorname{Tr}(\Pi A^2 \Pi)$$

Now using the fact that $\operatorname{Tr}(\Pi A^2 \Pi) = \operatorname{Tr}(A \Pi A)$, we can once again use the fact that $\Pi \preceq I$ to show that $A \Pi A \preceq A^2$ and hence $\operatorname{Tr}(A \Pi A) \leq \operatorname{Tr}(A^2)$. Putting things together, we have shown

$$\operatorname{Tr}(\Pi A \Pi \Pi A \Pi) \leq \operatorname{Tr}(A^2) \implies \|\Pi A \Pi\|_F^2 \leq \|A\|_F^2.$$

Thus, by summing the inequalities, $\Omega(\Pi A \Pi) \leq \Omega(A)$.

**3.**  The proof of (iii) is straightforward. The continuity of $\Omega$ comes from the fact that it is a norm on any finite dimensional Hilbert space. Moreover, since $\lambda_1 > 0$ or $\lambda_2 > 0$, $\Omega$ goes to infinity.  $\square$

**Remark 10.** *As in the previous sections, the fact that $\Omega$ satisfies these three properties is actually sufficient to complete the proof.*

Recall that $\mathcal{H}_n$ is the finite dimensional subset of $\mathcal{H}$ spanned by the $\phi(x_i)$. Recall that $\Pi_n$ is the orthogonal projection on $\mathcal{H}_n$, i.e.

$$\Pi_n \in \mathcal{S}(\mathcal{H}), \ \Pi_n^2 = \Pi_n, \ \operatorname{range}(\Pi_n) = \mathcal{H}_n.$$

Define $K_n^C(\mathcal{H})$ to be the following subspace of $K^C(\mathcal{H})$ :

$$K_n^C(\mathcal{H}) := \left\{ \Pi_n A \Pi_n \ : \ A \in K^C(\mathcal{H}) \right\}.$$

It is straightforward to show that $K_n^C(\mathcal{H}) \subset K^C(\mathcal{H})$ since projecting left and right preserves the linear inequalities.

**Proposition 9.** *Let $L$ be a lower semi-continuous function which is bounded below, and assume $\Omega$ satisfies Assumption 2. Then Eq. (15) has a solution $A^*$ which is in $K_n^C(\mathcal{H})$.*

*Proof.* In this proof, denote with $J$ the function defined by

$$\forall A \in \mathcal{S}(\mathcal{H})^p, \ J(A) := L(f_A(x_1), ..., f_A(x_n)) + \Omega(A).$$

Our goal is to prove that the problem $\inf_{A \in K^C(\mathcal{H})} J(A)$ has a solution which is in $K_n^C(\mathcal{H})$, i.e. of the form $\Pi_n A \Pi_n$ for some $A \in K_n^C(\mathcal{H})$.

**1.**  Let us start by fixing $A \in K^C(\mathcal{H})$.
First note that since $\Pi_n$ is the orthogonal projection on $\operatorname{span}(\phi(x_i))_{1 \leq i \leq n}$, in particular $\Pi_n \phi(x_i) = \phi(x_i)$ for all $1 \leq i \leq n$. Thus, for any $1 \leq i \leq n$,

$$f_A(x_i) = (\phi(x_i)^\top A_s \phi(x_i))_{1 \leq s \leq p} = (\phi(x_i)^\top \Pi_n A_s \Pi_n \phi(x_i))_{1 \leq s \leq p} = f_{\Pi_n A \Pi_n}(x_i).$$

Here, the first and last equalities come from the definition of $f_A$ and $f_{\Pi_n A \Pi_n}$. Thus,

$$J(A) = L(f_{\Pi_n A \Pi_n}(x_1), ..., f_{\Pi_n A \Pi_n}(x_n)) + \Omega(A).$$

Now since $\Omega$ satisfies Assumption 2, by the second point of Lemma 6, it holds $\Omega(\Pi_n A \Pi_n) \leq \Omega(A)$, hence

$$J(\Pi_n A \Pi_n) \leq J(A).$$

This last inequality combined with the fact that $K_n^C(\mathcal{H}) = \left\{ \Pi_n A \Pi_n \ : \ A \in K^C(\mathcal{H}) \right\} \subset K^C(\mathcal{H})$ show that

$$\inf_{A \in K_n^C(\mathcal{H})} J(A) = \inf_{K^C(\mathcal{H})} J(A).$$

**2.** Let us now show that $\inf_{A \in K_n^C(\mathcal{H})} J(A)$ has a solution. Let us exclude the case where $J = +\infty$, in which case $A = 0$ can be taken to be a solution.

Let $V_n$ be the injection $V_n : \mathcal{H}_n \hookrightarrow \mathcal{H}$. Note that $V_n V_n^* = \Pi_n$ and $V_n^* V_n = I_{\mathcal{H}_n}$. These simple facts easily show that

$$K_n^C(\mathcal{H}) = V_n K^C(\mathcal{H}_n) V_n^* = \left\{ V_n \tilde{A} V_n^* \ : \ \tilde{A} \in K_n^C(\mathcal{H}_n) \right\}.$$

Thus, our goal is to show that $\inf_{\tilde{A} \in K_n^C(\mathcal{H}_n)} J(V_n A V_n^*)$ has a solution.

By the first point of Lemma 6, since $V_n^* V_n = I_{\mathcal{H}_n}$, it holds

$$\forall \tilde{A} \in \mathcal{S}(\mathcal{H}_n), \ \Omega(V_n \tilde{A} V_n^*) = \Omega(\tilde{A}) \implies J(V_n \tilde{A} V_n^*) = L(f_{V_n \tilde{A} V_n^*}(x_1), ..., f_{V_n \tilde{A} V_n^*}(x_n)) + \Omega(\tilde{A}).$$

Let $\tilde{A}_0 \in K^C(\mathcal{H}_n)$ be a point such that $J_0 := J(V_n \tilde{A}_0 V_n^*) < \infty$. Let $c_0$ be a lower bound for $L$. By the third point of Lemma 6, there exists a radius $R_0$ such that for all $\tilde{A} \in \mathcal{S}(\mathcal{H}_n)$,

$$\|\tilde{A}\|_F > R_0 \implies \Omega(\tilde{A}) > J_0 - c_0.$$

Since $c_0$ is a lower bound for $L$, this implies

$$\inf_{\tilde{A} \in K^C(\mathcal{H}_n)} J(V_n \tilde{A} V_n^*) = \inf_{\tilde{A} \in K^C(\mathcal{H}_n), \, \|\tilde{A}\|_F \leq R_0} J(V_n \tilde{A} V_n^*).$$

Now since $L$ is lower semi-continuous, $\Omega$ is continuous by the last point of Lemma 6, and $\tilde{A} \mapsto (f_{V_n \tilde{A} V_n^*}(x_i))_{1 \leq i \leq n}$ is linear hence continuous, the mapping $A \mapsto J(V_n \tilde{A} V_n^*)$ is lower semi-continuous. Hence, it reaches its minimum on any non empty compact set. Since $\mathcal{H}_n$ is finite dimensional, the set $\left\{ \tilde{A} \in K^C(\mathcal{H}_n) \ : \ \|\tilde{A}\|_F \leq R_0 \right\}$ is compact (closed and bounded) and non empty (it contains $\tilde{A}_0$), and hence there exists $\tilde{A}_* \in K_n^C(\mathcal{H})$ such that $J(V_n \tilde{A}_* V_n^*) = \inf_{\tilde{A} \in K^C(\mathcal{H}_n), \, \|\tilde{A}\|_F \leq R_0} J(V_n \tilde{A} V_n^*)$. Going back up the previous equalities, this shows that $A_* := V_n \tilde{A}_* V_n^* \in K_n^C(\mathcal{H})$ and $J(A_*) = \inf_{A \succeq 0} J(A)$.

$\square$

**Lemma 7.** *The set $K_n^C(\mathcal{H})$ can be represented in the following way*

$$K_n^C(\mathcal{H}) = \left\{ (S_n^* \mathbf{B}_s S_n)_{1 \leq s \leq p} \in \mathcal{S}(\mathcal{H})^p \ : \ \mathbf{B} = (\mathbf{B}_s)_{1 \leq s \leq p} \in K^C(\mathbb{R}^n) \right\}$$

*In particular, for any $A \in K_n^C(\mathcal{H})$, there exists $p$ symmetric matrices $\mathbf{B} = (\mathbf{B}_s)_{1 \leq s \leq p} \in K^C(\mathbb{R}^n)$ such that*

$$\forall x \in \mathcal{X}, \ f_A(x) = \left( \sum_{1 \leq i,j \leq n} [\mathbf{B}_s]_{i,j} k(x_i, x) k(x_j, x) \right)_{1 \leq s \leq p}.$$

*Proof.* The proof is exactly analoguous to the proof of Lemma 3. $\square$

We will prove the following Thm. 13 which statement is that of Thm. 6 with more precise assumptions.

**Theorem 13.** *Let $L$ be lower semi-continuous and bounded below, and $\Omega$ satisfying Assumption 2. Then Eq. (5) has a solution of the form*

$$f_*(x) = \left( \sum_{i,j=1}^n [\mathbf{B}_s]_{i,j} k(x, x_i) k(x, x_j) \right)_{1 \leq s \leq p}, \qquad \text{for some family } \mathbf{B} = (\mathbf{B}_s)_{1 \leq s \leq p} \in K^C(\mathbb{R}^n).$$

*Moreover, if $L$ is convex, this solution is unique.*

*Proof of Thm. 13.* The proof is completely analoguous to that of Thm. 7, combining Lemma 7 and Proposition 9. $\square$

## C   Additional proofs

**Lemma 8.** *Let $\mathcal{X} \subset \mathbb{R}^d$, $d \in \mathbb{N}$, be a compact set with Lipschitz boundary. Let $m > d/2$. Then $W_2^m(\mathcal{X})$ is a multiplication algebra (see Definition 3).*

*Proof.* When $m \in \mathbb{N}$ and $m > d/2$, then $W_2^m(\mathbb{R}^d)$ is a multiplication algebra [AF03]. When $m \notin \mathbb{N}$, by Eq. 2.69 pag. 138 of [Tri06] we have that $F_{2,2}^m(\mathbb{R}^d)$ is a multiplication algebra when $m > d/2$, where $F_{2,2}^m$ is the Triebel-Lizorkin space of smoothness $m$ and order $2, 2$ and corresponds to $W_2^m(\mathbb{R}^d)$, i.e., $F_{2,2}^m(\mathbb{R}^d) = W_2^m(\mathbb{R}^d)$ [Tri06].

So far we have that $m > d/2$ implies that $W_2^m(\mathbb{R}^d)$ is a multiplication algebra, now we extend this result to $W_2^m(\mathcal{X})$. Note that since $\mathcal{X}$ is compact and with Lipschitz boundary, for any $f \in W_2^m(\mathcal{X})$ there exists an *extension* $\tilde{f} \in W_2^m(\mathbb{R}^d)$ such that $\tilde{f}|_\mathcal{X} = f$ and $\|\tilde{f}\|_{W_2^m(\mathbb{R}^d)} \leq C_1 \|f\|_{W_2^m(\mathcal{X})}$ with $C_1$ depending only on $m, d, \mathcal{X}$ (see Thm. 5.24 pag. 154 for $m \in \mathbb{N}$ and 7.69 when $m \notin \mathbb{N}$ pag 256 [AF03]). Then, since for any $f : \mathbb{R}^d \to \mathbb{R}$, by construction we have $\|f|_X\|_{W^m(\mathcal{X})} \leq \|f\|_{W^m(\mathbb{R}^d)}$ [AF03]. Then, for any $f, g \in W_2^m(\mathcal{X})$, denoting by $\tilde{f}, \tilde{g}$ the extensions of $f, g$, we have

$$\|f \cdot g\|_{W_2^m(\mathcal{X})} = \|\tilde{f}|_\mathcal{X} \cdot \tilde{g}|_\mathcal{X}\|_{W_2^m(\mathcal{X})} \leq \|\tilde{f} \cdot \tilde{g}\|_{W_2^m(\mathbb{R}^d)} \tag{31}$$

$$\leq C\|\tilde{f}\|_{W_2^m(\mathbb{R}^d)}\|\tilde{g}\|_{W_2^m(\mathbb{R}^d)} \leq CC_1^2\|f\|_{W_2^m(\mathcal{X})}\|g\|_{W_2^m(\mathcal{X})}. \tag{32}$$

To conclude $u : \mathcal{X} \to \mathbb{R}$ that maps $x \mapsto 1$ has bounded norm corresponding to $\|u\|_{W_2^m(\mathcal{X})}^2 = \int_\mathcal{X} dx$. So $W_2^m(\mathcal{X})$ when $m > d/2$ and $\mathcal{X}$ is compact with Lipschitz boundary is a multiplication algebra. $\qquad\square$

## D   Additional details on the other models

Recall that the goal is to solve a problem of the form Eq. (1), i.e.

$$\min_{f \in \mathcal{F}} L(f(x_1), ..., f(x_n)) + \Omega(f).$$

In this section, $\phi : \mathcal{X} \to \mathcal{H}$ will always denote a feature map, $k : \mathcal{X} \times \mathcal{X} \to \mathbb{R}$ a positive semi definite kernel on $\mathcal{X}$ ($k(x, x') = \phi(x)^\top \phi(x')$ if $k$ is the positive semi-definite kernel associated to $\phi$). Given a kernel $k$, $\mathbf{K} \in \mathbb{R}^{n \times n}$ will always denote the positive semi-definite kernel matrix with coefficients $\mathbf{K}_{i,j} = k(x_i, x_j)$, $1 \leq i, j \leq n$.

**Generalized linear models (GLM).** Consider generalized linear models of the form, $f_w(x) = \psi(w^\top \phi(x))$. Assume the regularizer is of the form $\Omega(f_w) = \frac{\lambda}{2}\|w\|^2$. Using the representer theorem [CL09], any solution to Eq. (1) is of the form $w = \sum_{i=1}^n \alpha_i \phi(x_i)$ and thus Eq. (1) becomes the following finite dimensional problem in $\alpha$:

$$\min_{\alpha \in \mathbb{R}^n} L(\psi(\mathbf{K}\alpha)) + \frac{\lambda}{2}\alpha^\top \mathbf{K}\alpha. \tag{33}$$

In the case where one wishes to learn a density function with respect to a basis measure $\nu$, a common choice of model is functions of the form

$$p_\alpha(x) = \frac{\exp(g(x))}{\int_{\tilde{x} \in \mathcal{X}} \exp(g(\tilde{x}))d\nu(\tilde{x})}, \qquad g(x) = \sum_{i=1}^n \alpha_i k(x_i, x).$$

where $k$ is a positive semi-definite kernel on $\mathcal{X}$. The prototypical problem one solves to find the best $p_\alpha$ is

$$\min_{\alpha \in \mathbb{R}^n} L(p_\alpha(x_1), ..., p_\alpha(x_n)) + \frac{\lambda}{2}\alpha^\top \mathbf{K}\alpha. \tag{34}$$

In the specific case where the loss function is the negative log likelihood $L(z_1, ..., z_n) = \frac{1}{n}\sum_{i=1}^{n} -\log(z_i)$, it can be shown that Eq. (34) is convex in $\alpha$.

In practice, we solve Eq. (33) by applying standard gradient descent with restarts, as the problem is non convex.

To solve Eq. (34), since the problem is convex, the algorithm is guaranteed to converge. However, since we can only estimate the quantity $\int_{\tilde{x}\in\mathcal{X}} \exp(g(\tilde{x}))d\nu(\tilde{x})$; we do so by taking a measure $\nu$ from which we can sample. However, this becomes intractable as the dimension grows, as the experiments on density estimation will put into light.

**Non-negative coefficients models (NCM).** Recall the definition of an NCM. It represent non-negative functions as $f_\alpha(x) = \sum_{i=1}^{n} \alpha_i k(x, x_i)$, with $\alpha_1, \ldots \alpha_n \geq 0$, given a kernel $k(x, x') \geq 0$ for any $x, x' \in \mathcal{X}$. In this case, the prototypical problem is of the form :

$$\min_{\alpha \geq 0} L(\mathbf{K}\alpha) + \frac{\lambda}{2}\alpha^\top\mathbf{K}\alpha. \tag{35}$$

If we are performing density estimation with respect to the measure $\nu$, one wishes to impose $\int_\mathcal{X} f_\alpha(x)d\nu(x) = 1$, which can be seen as an affine constraint over $\alpha$, since

$$\int_\mathcal{X} f_\alpha(x)d\nu(x) = \mathbf{u}^\top\alpha, \qquad \mathbf{u} = \left(\int_\mathcal{X} k(x, x_i)d\nu(x)\right)_{1\leq i\leq n} \in \mathbb{R}^n.$$

In this case, the prototypical problem will be of the form

$$\min_{\substack{\alpha \geq 0 \\ \mathbf{u}^\top\alpha=1}} L(\mathbf{K}\alpha) + \frac{\lambda}{2}\alpha^\top\mathbf{K}\alpha. \tag{36}$$

If $L$ is a convex smooth function, both problems Eq. (35) and Eq. (36) can be solved using projected gradient descent, since the projections on the set $\alpha \geq 0$ and the simplex $\{\alpha \in \mathbb{R}^n : \alpha \geq 0, \mathbf{u}^\top\alpha = 1\}$ can be computed in closed form.

In the main paper, we mention that NCM models do not satisfy **P2** i.e. that they cannot approximate any function arbitrarily well. We implement Example 2 in the following way. Let $g(x) = e^{-\|x\|^2/2}$. Take $k(x, x') = e^{-\|x-x'\|^2}$, $n$ points $(x_1, ..., x_n)$ taken uniformly in the interval $[-5, 5]$. To find the function $f_\alpha$ which best approximates $g$, we perform least squares regression, i.e. solve the prototypical problem Eq. (35) with the square loss function

$$L(y) = \frac{1}{2n}\sum_{i=1}^{n} |y_i - g(x_i)|^2.$$

We perform cross validation to select the value of $\lambda$ for each value of $n$. In Fig. 2, we show the obtained function $f_\alpha$ for $n = 100, 1000, 10000$. This clearly illustrates that with this model, we cannot approximate $g$ in a good way, no matter how many points $n$ we have.

**Partially non-negative linear models (PNM).** Consider partially non negative models of the form $f_w(x) = w^\top\phi(x)$, with $w \in \{w \in \mathcal{H} \mid w^\top\phi(x_1) \geq 0, \ldots, w^\top\phi(x_n) \geq 0\}$ (that is we impose $f_w(x_i) \geq 0$). Take $\Omega$ to be of the form $\frac{\lambda}{2}\|w\|^2$ in Eq. (1). Using the representer theorem in [CL09], we can show that there is a solution of this problem of the form $f_\alpha = \sum_{i=1}^{n} \alpha_i k(x, x_i)$, leading to the following optimization problem in $\alpha$ to recover the optimal solution:

$$\min_{\mathbf{K}\alpha \geq 0} L(\mathbf{K}\alpha) + \frac{\lambda}{2}\alpha^\top\mathbf{K}\alpha \tag{37}$$

If we want to impose that the resulting $f_\alpha$ sums to one for a given measure $\nu$ on $\mathcal{X}$, we proceed as in Eq. (36) and solve

$$\min_{\substack{\mathbf{K}\alpha \geq 0 \\ \mathbf{u}^\top\alpha=1}} L(\mathbf{K}\alpha) + \frac{\lambda}{2}\alpha^\top\mathbf{K}\alpha. \tag{38}$$

Figure 2: Best approximation of $g$ using NCM with *(left)* $n = 100$ *(center)* $n = 1000$ *(right)* $n = 10000$ points.

However, there is no guarantee that the resulting $f_\alpha$ will be a density, as will be made clear in the next section on density estimation.

In the experiments, we solve Eq. (37) and Eq. (38) in the following way. We first compute a cholesky factor of $\mathbf{K} : \mathbf{K} = \mathbf{V}^\top \mathbf{V}$. Changing variables by setting $\mathbf{V}\beta = \alpha$, the objective functions become strongly convex in $\beta$. We then compute the dual of these problems and apply a proximal algorithm like FISTA, since the proximal operator of $L$ is always known in our experiments.

# E  Additional details on the experiments

In this section, we provide additional details on the experiments. The code will be available online. Recall that we consider four different models for functions with non-negative outputs : GLM, PNM, NCM and our model.

**Kernels.**  All the models we consider depend on certain positive semi definite kernels $k$. In all the experiments, we have taken the kernels to be Gaussian kernels with width $\sigma$:

$$\forall x, x' \in \mathbb{R}^d, \ k(x, x') = \exp\left(-\frac{\|x - x'\|^2}{2\sigma^2}\right).$$

**Regularizers.**  For GLM, PNM and NCM, the regularizer for the underlying linear models are always of the form $\frac{\lambda}{2}\|w\|^2$ where $w$ is the parameter of the linear model, which translates to $\frac{\lambda}{2}\alpha^\top \mathbf{K}\alpha$ where the $\alpha$ are the coefficients of the finite dimensional representation. For our model, we always take the regularizer to be of the form $\lambda_1\|A\|_\star + \frac{\lambda_2}{2}\|A\|_F^2$.

**Parameter selection.**  In all experiments except for the one on density estimation in the main paper (in which we fix $\sigma = 1$ and select $\lambda$), we select the parameters $\sigma$ of the kernels involved as well as the parameters $\lambda$ for the regularizers using $K$ fold cross validation with $K = 5$. This means that once the data set has been generated, we randomly divide it into five sets, and train our model on 4 out of the 5 sets and test it on the remaining set, five times. We then train our model for the given $\sigma$, $\lambda$ or $\lambda_1, \lambda_2$, and report the performance on the test set, taking the mean of the $K$ performances to be the final performance. We also keep track of the standard deviation on these $K$ sets to avoid parameters which induce too big a variance. We then select the best parameters by doing a grid search. The code for this cross-validation is available online.

**Formulations and algorithms.**  The formulations of our three problems : density estimation, regression with Gaussian heteroscedastic errors, and multiple quantile regression, have been expressed in the main paper in a generic way involving functions with unconstrained outputs, and functions with outputs constrained to be non negative and sometimes summing to one. We always model functions with unconstrained outputs with a linear model with gaussian kernel, and model the functions with constrained outputs with the four models for non-negative functions we consider: ours, PNM, GLM and NCM.

In practice, we implement the methods PNM, GLM and NCM as explained in Appendix D. In particular, we use FISTA for PNM, and our model, dualizing the equality constraints for density estimation. This relies on the fact that the proximal operators of the log likelihood, the objective

Figure 3: Representation of the densities learned by the different models.

function for heteroscedastic regression as well as the pinball loss can be computed in closed form, and that the regularization is smooth in the right coordinates.

**Details on the experiments of the main text.**    Here, we add a few precisions on the toy distributions we have used to sample data and the number of sampled used when not specified in the main text.

- For heteroscedastic regression, the data was generated as the toy data in section 5 of [LSC05], with $n = 80$ points.

- For quantile regression, the data points $(x_i, y_i)$ were generated according to the following distribution for $(X, Y) : X \sim \frac{1}{2}U(0, 1/3) + \frac{1}{2}U(2/3, 1)$ and $Y|x \sim \mathcal{N}(0, \sigma(x))$ where

$$\sigma(x) = \begin{cases} -x + 1/3 & \text{for } 0 \leq x \leq 1/3 \\ x - 2/3 & \text{for } 2/3 \leq x \leq 1 \; . \\ 0 & \text{otherwise} \; . \end{cases}$$

Here, $U$ stands for the uniform distribution. Moreover, in order to perform the experiments in the main paper, we have used 500 sample points.

**Density estimation in dimension** 10 **with** $n = 1000$**.**    In this paragraph, we consider the following experiment. Let $d = 10$, $X \in \mathbb{R}^d$ be a random variable distributed as a mixture of Gaussians :

$$X \sim \frac{1}{2}\mathcal{N}(-2e_1, 1/\sqrt{2\pi}I_d) + \frac{1}{2}\mathcal{N}(2e_1, 1/\sqrt{2\pi}I_d)$$

where $e_1$ is the first vector of the canonical basis of $\mathbb{R}^d$.

Let $n = 1000$ and let $(x_1, ..., x_n)$ be $n$ iid samples of $X$. We perform the four different methods, cross validating both the regularization parameter $\lambda$ and the kernel parameter $\sigma$ at each time. We learn the density in the form

$$p(x) = f(x)\nu(x), \qquad \nu \text{ is the density associated with } \mathcal{N}(0, 5I_d).$$

We then use our models for densities to compute the best $f$ in its class using the negative log-likelihood as a loss function. It is crucial that we can sample from $\nu$ in order to approximate the integral in the case of GLMs.

In order to visualize the results of the different algorithms in Fig. 3, we compute the learnt distribution $p$, and then sample randomly $n_0 = 500$ points from a uniform distribution on the box centered at $0$ and of width $5$ in order to explore regions where the density is close to zero, $n_0$ points sampled from the true distribution of the data, in order to explore points where the density is representative, and $n_0$ points on the line $[-4, 4] \times \{0\}^{d-1}$ where the density is at its highest. We then project onto the first coordinate, i.e. given a point $x = (x_i)_{1 \leq i \leq d}$ and the associated predicted density $p(x)$, we plot the point $(x_1, p(x))$. Note that for readability, we have used the same scale for our model and the PNM, and a smaller scale for the two others since the learnt density is much flatter.

Let us now analyse the results in Fig. 3. Note that in terms of performance, i.e. log likelihood on the test set, the first two models (PNM and our model) are quite close and are better than the two others.

- **PNM**. As in $d = 1$ we see that for $d = 10$ the problems of non-negativity for PNM are exacerbated, making it not suitable to learn a probability distribution. Indeed there are low density regions where the optimization problem pushes the model to be negative. Since by constraint we have $\int f d\nu = 1$, the volume of the negative regions is used to push up the function in the regions with high density. So $\int |f| d\nu \gg 1$, while it should be $\int |f| d\nu = 1$. This is confirmed by the behavior of the cross validation.

- **Our model** Our model seems to perform reasonably well.

- **NCM**. This problem is rather difficult for NCM. Indeed, as the width of the kernel decreases, the model is unable to learn since it overfits in the direction $e_1$ and it would require way more points than $n = 1000$. However, as soon as the width of the kernel is good for $e_1$, the learnt distribution becomes too heavy tailed in the direction orthogonal to $e_1$.

- **GLM**. It is interesting to note that GLM completely fails, because the measure $\nu$ which we take as a reference measure has a support which has only double variance compared to $p$, but in 10 dimensions it corresponds to a support with way larger volume compared to the one of the target distribution. In particular, the estimation of the integral, which was possible in $d = 1$ with 10000 i.i.d. points from $\nu$, in 10 dimensions becomes almost impossible (it would require way more sampling points). Note that we sample the points from $\nu$ to simulate the real-world situation where $p$ is a measure from which it is difficult to sample from, while $\nu$ is an simple measure to sample from which contains the support of $p$. Further experiments show that if one takes the target distribution to sample, one obtains a good model, which reassures us in the fact that this is not a coding error but a real phenomenon.

# F    Relationship to [4]

As mentioned in the main paper, the model in Eq. (4) has already been considered in [4] with a similar goal as ours. This paper is a workshop publication that has only be lightly peer-reviewed and contains fundamental flaws. In particular, they provide an incorrect characterization of the solution of Eq. (5), that limits the representation power of the model to the one of non-negative coefficients models, that, as we have seen in Sec. 2.1 and in Example 2, has poor approximation properties and cannot be universal. This severe limitation affects also the optimization framework (which also only relies on general-purpose toolboxes such as CVX (http://cvxr.com/cvx/), which are not scalable to large $n$).

Indeed, in their main result, the representer theorem incorrectly characterizes $A^*$ the solution of Eq. (5) as

$$A^* \in R_n \cap \mathcal{S}(\mathcal{H})_+, \qquad R_n = \left\{ \sum_{i=1}^n \alpha_i \phi(x_i) \otimes \phi(x_i) \mid \alpha \in \mathbb{R}^n \right\},$$

and $\mathcal{S}(\mathcal{H})_+ = \{ A \in \mathcal{S}(\mathcal{H}) \mid A \succeq 0 \}$. Note, however that $R_n \subseteq \mathcal{S}(\mathcal{H}_n) \subset \mathcal{S}(\mathcal{H})$ by construction, where $\mathcal{H}_n = \mathrm{span}\{\phi(x_1), \dots, \phi(x_n)\}$. So their characterization corresponds to

$$A^* \in \left\{ A = \sum_{i=1}^n \alpha_i \phi(x_i) \otimes \phi(x_i) \mid \alpha \in \mathbb{R}^n, A \succeq 0 \right\}.$$

Now, for simplicity, consider the interesting case where $\phi$ is universal and $x_1, \dots, x_n$ are distinct points. Then $(\phi(x_i))_{i=1}^n$ forms a basis for $\mathcal{H}_n$ and the only $\alpha_1, \dots, \alpha_n \in \mathbb{R}$ that guarantee $A \succeq 0$ are $\alpha_1 \geq 0, \dots, \alpha_n \geq 0$, i.e.,

$$R_n = \left\{ A = \sum_{i=1}^n \alpha_i \phi(x_i) \otimes \phi(x_i) \mid \alpha_1 \geq 0, \dots, \alpha_n \geq 0 \right\}.$$

Note that this class of operators leads only to *non-negative coefficients models*. Indeed, let $A \in R_n$ and denote by $k(x, x')$ the function $k(x, x') = (\phi(x)^\top \phi(x'))^2$, then

$$f_A(x) = \phi(x)^\top A \phi(x) = \sum_{i=1}^n \alpha_i (\phi(x)^\top \phi(x_i))^2 = \sum_{i=1}^n \alpha_i k(x, x_i), \quad \forall \, x \in \mathcal{X}.$$

Since $k$ is a kernel (it is an integer power of $\phi(x)^\top \phi(x')$ that is a kernel [SS02]) and $\alpha_1 \geq 0, \dots, \alpha_n \geq 0$, then $f_A$ belongs to the non-negative coefficients models.

Instead, we know by our Thm. 1 that $A^* \in \mathcal{S}(\mathcal{H}_n)_+$ and more explicitly, by Thm. 2 that $A^*$, the solution of Eq. (5) is characterized by the non-positive part operator of a symmetric matrix $[\cdot]_+$. By Thm. 3 we already know that our model is universal while NCM and thus the characterization in [4] cannot be universal.