[Reviews · NeurIPS 2020]

Review 1

Summary and Contributions: In this paper, statistical models for non-negative functions are proposed. The basic idea is to use quadratic forms with positive semi-definite matrices. The authors proved the proposed models have some favorable properties including convexity, representer theorem, universal approximation, and complexity bound like the standard kernel methods. Numerical experiments are demonstrated to show the effectiveness of the proposed modeling.

Strengths: Differently from the standard transformation using the exponential function or the cut-off function, the proposed models have some favorable properties such as convexity, representer theorem, universal approximation, and complexity bound like the standard kernel methods. The dual expression of the optimizaiton problem leads to a simple optimization algorithm.

Weaknesses: The computational complexity seems demanding for massive data. The authors suggested that approximation techniques such as random features or Nystrom approximation. It would be more informative if numerical results using some approximation methods for huge data are reported. All the numerical experiments in section 6, the target function is defined on the 1-dim real line. In my understanding, the proposed method is applicable to the estimation of functions on the multi-dimensional space. Showing the computational cost and the prediction accuracy on such multi-dimensional problems will be beneficial.

Correctness: The mathematical part seems to be correct.

Clarity: The paper is clearly written.

Relation to Prior Work: Some relation to prior works was investigated.

Reproducibility: Yes

Additional Feedback: Thanks for the response file. I keep my score as this is a very interesting paper.


Review 2

Summary and Contributions: The function is modeled as a quadratic form of a feature map \phi, parametrized by a positive semi-definite operator A. This is optimized subject to nuclear and Frobenius norm penalties. Under fairly general losses, Theorem 1 provides a finite (O(n^2)) dimensional solution to this problem. The paper then provides a dual formulation in terms of O(n) parameters, and shows that, for suitable kernels, the model is dense in the space of non-negative continuous functions. Section 5 extends the model to the cases of linear and convex conical constraints. Finally, some numerical experiments are presented, comparing behavior of the proposed model to other approaches for density estimation, heterscedastic regression, and quantile regression.

Strengths: The authors propose a reasonable method for an important and (perhaps surpisingly) challenging problem.

Weaknesses: I believe that there are some issues in the presentation of the proposed and competing methods, and in the empirical comparisons. Details are below.

Correctness: I feel that there are some significant issues in the comparison with NCM. Details are below.

Clarity: Language-wise, the paper is well written, but I think the approach could be a bit better motivated, as I describe in detail below.

Relation to Prior Work: 1) Intuitively, the proposed model seems hugely over-parametrized (O(n^2) parameters!) for the described purpose of modeling non-negative functions. Indeed, in the proof of Theorem 3, to obtain a cc-universal approximator, it suffices to take an operator A of the form A = ww^T. From a statistical perspective, a preferable model would simply be f_w(x) = (w^T \phi(x))^2. The benefit of allowing A to be full-rank is convexity, which makes the model easier to fit. The prior knowledge that the optimization problem has an exact rank-1 solution is presumably the motivation for imposing a nuclear norm constraint. I think clarifying this logic would help motivate the model, as well as the elastic net regularization proposed in (6). 2) I may be missing something here, but the comparison of the proposed model with the "non-negative coefficients model" (NCM) in this paper seems very strange to me, for a several related reasons: a) On the theoretical side, the criticism of NCM methods is that, with a fixed bandwidth and kernel, they cannot approximate arbitrary continuous functions. I am confused about why one would fix the bandwidth. Typically, one sends the bandwidth to 0 as n -> infinity, and, in this case, NCM methods *can* approximate arbitrary continuous functions. b) Relatedly, in supplemental the Density Estimation experiment, it is mentioned that \sigma = 1, whereas \lambda is selected by cross-validation. It seems obvious to me that this favors the method that is sensitive to \lambda. c) In the 10-dimensional density estimation experiment in the Appendix, where \sigma is also selected by cross-validation and the comparison seems more fair, no quantitative or otherwise clear results are presented. The discussion suggests that the optimal bandwidth in the direction e_1 is different from the optimal bandwidth in other directions, but I don't find this plausible because the optimal bandwidth here should depend mostly on the covariances of the Gaussian mixture components (which are isotropic) rather than the means. The distribution learned by NCM (displayed in Figure 3) simply seems highly over-smoothed. d) The universal approximation result effectively shows that the model is asymptotically unbiased, but variance is not discussed. This seems like an important omission, especially since the model has far more parameters (O(n^2)) than NCM methods (O(n)). The experiments similarly address bias but not variance, since only a single trial is reported. In short, while I think the proposed method may offer significant advantages over NCM, I don't think the paper convincingly explains what these advantages are. Since NCM is probably the method most closely related to the proposed method and would be a go-to method for most of the nonparametric problems discussed in this paper, I think it's important to communicate this point correctly. I leave it to the authors to decide how to do this, but one way would be by comparison to the rank-1 model described above.

Reproducibility: No

Additional Feedback: To summarize my review, while I think the paper has some good ideas, I think the paper needs to do two main things to be acceptable: 1) Correct the presentation of NCM methods and clarify the relationship between the proposed method and NCM methods. 2) Provide more comprehensive, quantitative experimental results that a) capture both bias and variance of each method, especially in higher dimensions, b) include aggregate results over multiple IID replicates, and c) more fairly tune the hyperparameters of the reported methods, and report the ranges of hyperparameters used (or, even better include code for reproducing the experiments). -----------AFTER READING THE REBUTTAL----------- Thanks to the authors for their detailed response. I think the clarifications and proposed changes by the authors clear up my main issues with the paper. I've increased my score significantly. Minor comments: - "kernel density estimation... it can’t approximate a density with i.i.d. samples faster than n^{−1/(d+1)} even if the density is arbitrarily smooth": Actually, a rate of n^{−2/(d+2)} (in L2 risk) should be possible, since a second-order kernels can be non-negative (i.e., up to 2 orders of smoothness can be leveraged with a symmetric, non-negative kernel). This is a small detail, and, indeed, the theoretical convergence rate of kernel density estimation (KDE) with non-negative kernels is poor. But, this slow convergence rate is quite different from saying that NCM cannot approximate less smooth functions whatsoever, as the paper initially claimed... - "the best parameters found by cross-validation for NCM are σ= 0.5 and λ= 10−4. Since the value of σ is not on the left extreme of the range": Thank you for verifying this. It reduces my concerns somewhat.


Review 3

Summary and Contributions: A mathematical framework is developed for dealing with problems involving nonnegative functions, such as density estimation. A representer theorem is given for regularization problems, reducing infinite-dimensional problems involving nonnegative functions to finite dimensional ones. The methodology is extended to the case that a function lies in a convex cone. Comparison is done with three alternative methods: nonnegative coefficient models, partially nonnegative linear models, and generalized linear models (GLMs). The first two are less interesting perhaps, but the GLM is a natural method of dealing with nonnegative functions. However, it's computational complexity, eg for density estimation in high-dimensions, may become prohibitive. In contrast, the proposed methodology can be used in high dimensions, and gives very good performance.

Strengths: This seems to be an original work, it is mathematically sound, and computationally feasible. The idea is simple, which is good, namely the point evaluator of a nonnegative function is represented as a quadratic form, which is necessarily nonnegative. Thus, each nonnegative function in a nonnegative function space is represented by a positive semi-definite matrix/operator. This cleverly gives a mathematical handle which we can work with (computationally and theoretically) more easily than otherwise.

Weaknesses: Essentially as I understand it the problem of estimating a non-negative function is replaced by the problem of estimating a non-negative definite matrix/operator. So I'd say this is not quite as simple as estimating linear models, ie the proposed model does not benefit from all the "good" properties of the linear model. Perhaps something can be said on how much estimation complexity this adds.

Correctness: Yes.

Clarity: Yes

Relation to Prior Work: Yes.

Reproducibility: Yes

Additional Feedback: Response to reviewer feedback: I thank the authors for their response. In terms of complexity, the learning rate may be the same as for linear models, but the necessity to estimate a positive definite operator surely adds some algorithmic complexity? My overall score lowered slightly not because of the response but because I now realize the acceptance rate is only 20%. For a top score, some more theoretical development of the function spaces of non-negative functions would have been interesting.


Review 4

Summary and Contributions: This paper is on the topic of learning functions that output non-negative values. Applications span density estimation, quantile and isotonic regression. The basic model studied is a positive-definite quadratic form after feature space lifting of the data. The paper shows that the implied learning problem with convex losses is convex, and admits a Representer Theorem which allows finite dimensional optimization even if the feature space is infinite dimensional. If the feature space is universal, the paper shows that the capacity of the model covers all non-negative functions. Experiments show convincing improvement over more heuristic baselines.

Strengths: This paper is very well written and provides a useful context for the study of learning non-negative functions, which arise in many applications. Many natural questions are answered, e.g., reduction to finite dimensional optimization via Representer Theorems, approximation theory, and limitations of other heuristic approaches. Results are provided for 3 different ML applications.

Weaknesses: The relationship to results of reference [4], which is very close in spirit, is not properly explained. Only 2d small toy datasets are used in the experiments. Section 5 on interesting extensions is very short and bit hasty.

Correctness: Yes.

Clarity: Yes.

Relation to Prior Work: Yes, except for relationship to [4]. The paper would also benefit from a short review of Sum-of-Squares programming which centers on positive polynomials.

Reproducibility: Yes

Additional Feedback:

[Author Response · NeurIPS 2020]

Thanks to the reviewers for their constructive comments.

**Reviewer #1.** Thanks for the positive feedback. We agree that a multidimensional experiment would be beneficial for
the main paper. Note that the paper already has a multidimensional experiment in Appendix E, that will be moved at the
end of the experimental section. Moreover we will add some details on the computational cost and fast algorithms to
solve the dual problem in Thm. 2 to the end of Sect. 3 (they are now at the end of Appendix B.5).

**Reviewer #2.** We thank the reviewer for the careful reading and thoughtful comments.

1) As correctly pointed out by Rev2, a model of the form $f_w(x) = (\phi(x)^\top w)^2$ is sufficient for Theorem 3 (and also for
Theorem 4). Such formulation would lead to non-convex optimization problems as already noticed by the reviewer.
Of course if convexity is not an issue, it is possible to restrict the model to rank-1, keeping all the results of Section
4. However, surprisingly, note that the dual formulation of the problem (Thm. 2) shows that the matrix $A$ also admits
a representation in terms of only $n$ degrees of freedom, instead of $n^2$ (given $n$ coefficients $\alpha$, the operator $A$ can be
recovered via Eq. 11 and then Eq. 7). Thus using the whole matrix $A$ allows on the one hand to still have a representation
in terms of $n$ degrees of freedom. On the other hand it leads to a convex problem and allows to control the rank of $A$
via elastic-net regularization. To conclude, we agree that the question raised by Rev2 can be useful to better understand
the value of the proposed approach. So we will add all the reasoning above as a remark right after Thm. 2.

2) We would like to point out that we study explicitly the variance of the proposed method in Theorem 5 where we
bound the Rademacher complexity of the proposed estimator. Indeed, by using a standard argument based on the
Rademacher complexity (see [29] Chapter 26, or [3] paragraph 4.5 and in particular Eq. 13) we can derive the following
learning rate. Let the population risk be defined as $R(f) = \mathbb{E}_{x,y} \ell(y, f(x))$ for some $G$-Lipschitz loss function $\ell$ and
$\widehat{R}_D$ be the empirical version $\widehat{R}_D(f) = \frac{1}{n} \sum_{i=1}^n \ell(y_i, f(x_i))$ for a given dataset $D$ of $n$ examples. Given a norm $\|\cdot\|_\circ$
(e.g., Frobenius or nuclear), a feature map $\phi$ and a radius $L$, define the class of estimators $\mathcal{F}_{\phi,L}^\circ = \{f_A \mid \|A\|_\circ \leq L\}$.
Denote by $\widehat{f}_{D,L} = \arg\min_{f \in \mathcal{F}_{\phi,L}^\circ} \widehat{R}_D(f)$ the empirical risk minimization solution over the set $\mathcal{F}_{\phi,L}^\circ$, then

$$\mathbb{E}_D R(\widehat{f}_{D,L}) \leq \inf_{f \in \mathcal{F}_{\phi,L}^\circ} R(f) + 2\,\mathbb{E}_D\Big[ \sup_{f \in \mathcal{F}_{\phi,L}^\circ} |R(f) - \widehat{R}_D(f)| \Big] \leq \inf_{f \in \mathcal{F}_{\phi,L}^\circ} R(f) + 2G\,\mathcal{R}_n(\mathcal{F}_{\phi,L}^\circ),$$

where $\mathcal{R}_n(\mathcal{F}_{\phi,L}^\circ)$ is the Rademacher complexity of the set $\mathcal{F}_{\phi,L}^\circ$ and is bounded by $O(Lc^2/\sqrt{n})$ by Theorem 5 ($c$
is the bounding constant of the kernel, i.e., $c = \sup_{x \in \mathcal{X}} \|\phi(x)\|$). Now, assuming that there exists an operator
$A_\star$ with $\|A_\star\|_\circ$ finite (in particular it could be rank-1, i.e. $A_\star = w_\star w_\star^\top$ for some $w_\star$), such that the learning
problem is well posed, i.e. $\inf_{f \in C(X)} R(f) = R(f_{A_\star})$, and choosing $L = \|A_\star\|_\circ$, we obtain the learning rate
$\mathbb{E}_D R(\widehat{f}_{D,L}) - R(f_{A_\star}) = O(c^2 G \|A_\star\|_\circ/\sqrt{n})$, that is comparable to the one of kernel linear models [29]. We will add
this paragraph as a discussion after Thm. 5, to better clarify the variance and learning rates for the proposed approach.

3) The NCM approach, i.e., approximating a function with non-negative combination of non-negative kernel functions
is employed usually in kernel density estimation methods. This model is well known to be quite rigid when the kernel
function is non-negative, indeed it can't approximate a density with i.i.d. samples faster than $n^{-1/(d+1)}$ even if the
density is arbitrarily smooth (see, e.g., [33]). This happen also when the density is of the form $f_\star(x) = e^{-V(x)}$
with $V(x)$ an infinitely smooth potential. Instead, in this case, according to the Point 2) above, the proposed method
has a faster learning rate $O(\|w_\star\|^2/\sqrt{n})$, where $\phi$ is the Sobolev kernel, since by Thm. 4 there exists a $w_\star$ s. t.
$f_\star(x) = (w_\star^\top \phi(x))^2$. We will add this example in Section 4 to clarify the difference between NCM and our method.

4) As suggested by the reviewer we will add a quantitative version of the experiments in the appendix, with 50 i.i.d.
repetitions and the resulting error bars. In Appendix E are reported many experimental details. We will add in the main
text a more detailed description on how we performed cross validation on both $\sigma \in [10^{-3}, 10^3]$ and $\lambda \in \{0\} \cup [10^{-8}, 1]$
(logarithmic scale, 20 intervals). We would like to note that in the multivariate experiment ($d = 10$), in the appendix, the
best parameters found by cross-validation for NCM are $\sigma = 0.5$ and $\lambda = 10^{-4}$. Since the value of $\sigma$ is not on the left
extreme of the range we would exclude that the result obtained by NCM in this experiment was due to oversmoothing.
We instead interpret such result in the light of the different learning rates $O(n^{-1/11})$ for NCM and $O(n^{-1/2})$ for our
method. In any case we will repeat the experiment (with 50 repetitions) and on a range $\sigma \in [10^{-10}, 10^{10}]$ (log scale
with 100 steps). To conclude we would like to recall that we will publish the code on GitHub (python + scipy).

**Reviewer #3.** We thank the reviewer for the positive feedback. As recalled in the Point 1) and 2) above, the model can
be expressed in terms of $n$ degrees of freedom via duality in Thm. 2. In terms of statistical complexity we achieve a
learning rate that is similar to the one for linear models (see Point 2). We will add these discussions after Thm. 2 and 5.
More details on the computational complexity of the dual formulation (Thm. 2) are at the end of Appendix B.5.

**Reviewer #4.** Thanks for the thoughtful comments. An extensive explanation about the relation with [4] and its intrinsic
limitations is already reported in Appendix F. We will move part of the content in Section 2, where we will also add a
short review about main result on SoS programming.

[Meta-Review · NeurIPS 2020]

The focus of the work is modeling non-negative functions (or more generally functions with output in convex cones). The authors propose a non-parametric approach to tackle this task. Particularly, they approximate non-negative functions by quadratic forms in reproducing kernel Hilbert spaces parameterized by positive semi-definite operators. They show that the proposed approach have various favorable properties including convexity, universal approximation, representer theorem (hence it is computationally tractable) and it gives rise to complexity bounds alike to standard kernel approaches. The efficiency of the approach is illustrated in density estimation, heteroscedastic Gaussian process estimation and multiple quantile regression. This a nice submission: the paper is clearly written, it delivers both interesting theoretical insights and have practical relevance.